**The Impact of Increasing Stratospheric Radiative Damping on the QBO Period**
Tiehan Zhou[1,2], Kevin DallaSanta[1,3], Larissa Nazarenko[1,2], Gavin A. Schmidt[1], Zhonghai Jin[1]
[1]NASA Goddard Institute for Space Studies, New York, NY
[2]Center for Climate Systems Research, Columbia University, New York, NY
[3]Universities Space Research Association, Columbia, MD
Correspondence to: Tiehan Zhou (tz2131@columbia.edu)
**Abstract.** Stratospheric radiative damping increases as atmospheric carbon dioxide concentration rises.
We use the one-dimensional mechanistic models of the QBO to conduct sensitivity experiments and
find that the simulated QBO period shortens due to the enhancing of radiative damping in the
stratosphere. This result suggests that increasing stratospheric radiative damping due to rising $CO_2$ may
play a role in determining the QBO period in a warming climate along with wave momentum flux
entering the stratosphere and tropical vertical residual velocity, both of which also respond to
increasing $CO_2$.
**1. Introduction**

18       The quasi-biennial oscillation (QBO) dominates the variability of the equatorial middle and lower

stratosphere and is characterized by a downward propagating zonal wind regime that regularly changes
from westerlies to easterlies. The QBO period ranges from 22 to 34 months with its average being
slightly longer than 28 months. The QBO not only manifests itself in the equatorial zonal winds, but also
leaves an imprint on the temperature in both the tropics and extratropics (Baldwin et al., 2001 and
references therein).
The QBO has far-reaching implications for global weather and climate systems. First of all, the QBO
exerts a marked influence on the distribution and transport of various chemical constituents such as
ozone ($O_3$) (e.g., Hasebe, 1994), water vapor ($H_2O$) (e.g., Kawatani et al., 2014), methane ($CH_4$), nitrous
oxide ($N_2O$), hydrogen fluoride (HF), hydrochloric acid (HC1), odd nitrogen species ($NO_y$) (e.g.,
Zawodny and McCormick, 1991), and volcanic aerosol (Trepte and Hitchman, 1992). Secondly, it is
well appreciated that the QBO influences the extratropical circulation in the winter stratosphere, which
is commonly known as the Holton–Tan effect (Holton and Tan, 1980; Labitzke, 1982). It has been noted
that the effect of the QBO on the extratropical winter stratosphere impacts the severity of stratospheric
ozone depletion (e.g., Lait et al., 1989). Furthermore, taking account of the QBO improves the simulation
and predictability of the extratropical troposphere (e.g., Marshall and Scaife, 2009). Finally, through its
modulation of temperature and vertical wind shear in the vicinity of the tropical tropopause, the QBO
influences tropical moist convection (Collimore et al., 2003; Liess and Geller, 2012), the El Niño-
Southern Oscillation (ENSO) (Gray et al., 1992; Huang et al., 2012; Hansen et al. 2016), the Hadley
circulation (Hitchman and Huesmann, 2009), the tropospheric subtropical jet (Garfinkel and Hartmann,
2011a, 2011b), the boreal summer monsoon (Giorgetta et al., 1999), and the Madden-Julian Oscillation
(Yoo and Son, 2016). Intriguingly, the QBO is also reported to influence the activities of tropical
cyclones (Gray et al., 1984; Ho et al., 2009), albeit this issue is still unsettled (Camargo and Sobel, 2010)
and needs further study.
Efforts to understand and simulate the QBO have been ongoing ever since its discovery by Ebdon
(1960) and Reed et al. (1961). Lindzen and Holton (1968) and Holton and Lindzen (1972) developed
the classical theory of the QBO. Namely, as waves propagate upward, they are attenuated by thermal
damping, encounter critical levels, and accelerate and decelerate the mean flow, providing momentum
sources for both the westerly and easterly phases of the QBO.
Holton and Lindzen's (1972) model (hereafter referred to as HL model) was further simplified by
Plumb (1977), the elegance of which made it a standard paradigm for the QBO. In Plumb's (1977)
Boussinesq formulation, the QBO period is inversely dependent upon both the momentum flux and
thermal dissipation rate. Hamilton (1981) further highlighted the role of the radiative damping rate on
both the realistic vertical structure and the realistic period of the QBO.
By adopting higher vertical resolutions and incorporating various gravity wave parameterization
schemes, many state-of-the-art climate models have shown the capability to self-consistently simulate
the QBO (Scaife et al., 2000; Giorgetta et al., 2002, 2006; Rind et al., 2014, 2020; Geller et al., 2016a;
Richter et al., 2020a, 2020b). Given the important implications of the QBO for the global climate system,
it is natural to ask how the QBO will change in a warming climate.
Giorgetta and Doege (2005) showed a shortening of the QBO period in their doubled $CO_2$
experiments. They reasoned that both the weakening of the tropical upwelling and the prescribed
increase of gravity wave sources lead to the reduction of the QBO period in a warming climate. However,
most climate models project a strengthening rather than weakening of tropical upwelling in a warmer
climate (Butchart et al., 2006; Butchart 2014; Li et al., 2008). Employing a model without any
parametrized non-orographic gravity waves, Kawatani et al. (2011) demonstrated that the intensifying
tropical upwelling in a warming climate dominates the counteracting effect of enhanced wave fluxes and
consequently projected a lengthening of the QBO period. Using fixed sources of parametrized gravity
waves, Watanabe and Kawatani (2012) also projected an elongation of the QBO period in a warming
climate and ascribed it to the stronger tropical upwelling. Analyzing four Coupled Model
Intercomparison Project phase 5 (CMIP5) models that could simulate a reasonable QBO, Kawatani and
Hamilton (2013) found that the projected trends of the QBO period were inconsistent in sign. They
further investigated the 60-year operational balloon-borne radiosonde observations provided by the Free
Berlin University and detected no significant trend in the QBO period. Richter et al. (2020b) investigated
the response of the QBO to doubled and quadrupled $CO_2$ climates among eleven models that participated
in Phase 1 of the Stratospheric-tropospheric Processes And their Role in Climate QBO-initiative (QBOi;
Butchart et al., 2018), and found no consensus on how the QBO period would respond to a changing
climate. Recently, Butchart et al. (2020) evaluated ten Coupled Model Intercomparison Project phase 6
(CMIP6) models with realistic QBO in two Shared Socioeconomic Pathways (SSPs, Gidden et al., 2019)
scenario simulations and surprisingly found that the QBO period shortens in seven of those ten models
in both in both SSP3-7.0 and SSP5-8.5 scenarios although only two and three models show a significant
shortening trend in the respective scenarios.

It is challenging to ascertain the trend of the QBO period in a warming climate. On one hand, a

speeding-up of the Brewer-Dobson circulation in a warming climate leads to a lengthening of the QBO
period in most climate models. On the other hand, there is a robust increase in the vertical component of
the EP flux for both eastward and westward propagating waves (Richter et al., 2020b; Butchart et al.,
2020), indicating that the QBO period shortens due to the enhanced wave driving in a warming climate.
The competing effects between enhanced wave driving and a faster Brewer-Dobson circulation suggests
that trends in the QBO period are likely to be small and difficult to detect due to the large cycle-to-cycle
variability that is reproduced by climate models (Butchart et al., 2020). In addition, uncertainty in the
representation of the parameterized gravity waves make it more elusive to detect the trend of the QBO
period in a warming climate (Schirber et al., 2015; Richter et al., 2020b).

Given the fact that the QBO period is influenced by the radiative damping (Plumb 1977; Hamilton

1981), a natural question to ask is whether it could play a role on the trend of the QBO in a warming
climate. Fels (1985) estimated that the radiative damping time under a doubling of $CO_2$ would decrease
by about 23%. His estimate implies a shortening of the QBO period as the radiative damping rate
increases.
It is well-known that enhanced wave fluxes entering the stratosphere and stronger tropical upwelling
individually play a dominant role in determining the trends in the QBO period in a warming climate.
Does the competing effect between them leave some room for increasing stratospheric radiative damping
to exert an influence on the QBO period? In this paper, we use the HL model to isolate the effect of
radiative damping on the QBO period by assuming that the momentum flux entering the stratosphere
doesn't change in our experiments. Observational and modeling studies (Andrews et al., 1987; Kawatani
et al., 2009, 2010, 2011; Richter et al., 2020b; Holt et al., 2020) showed that the wave forcing spectrum
is similar to a discrete two-wave spectrum rather than red-noise or white-noise, all of which are
illustrated in Saravanan (1990). Accordingly, the QBO is indeed sensitive to stratospheric radiative
damping, and the HL model is suitable for us to conduct the sensitivity analysis.
The remainder of this paper is organized as follows. Section 2 investigates the sensitivity of the QBO
period to the radiative damping using HL's original model. Section 3 explores the sensitivity of the QBO
period to the radiative damping using a modified HL model where the semiannual forcing is removed.
Discussion and conclusions are presented in Sections 4 and 5 respectively.

**2. Sensitivity of the QBO period to enhanced stratospheric radiative damping in the original HL**
**model**
In the HL model the governing equation of mean flow emerges after the primitive momentum
equation is meridionally averaged over some suitable latitudinal belt over the equator.
$$\frac{\partial \overline{u}}{\partial t} = -\frac{1}{\rho_0}\frac{\partial}{\partial z}\left[\sum_{i=0}^{1}\overline{F_i}\right] + K_z\frac{\partial^2 \overline{u}}{\partial z^2} + G \tag{1}$$

where $\overline{u}$ is mean zonal wind, $\rho_0$ is mean density, $\overline{F_i}$ is the meridionally averaged vertical Eliassen-Palm
flux associated with wave $i$, the index $i$ refers to the individual waves, $K_z$ is a vertical eddy diffusion
coefficient, $t$ is time, $z$ is altitude, and $G$ is semiannual forcing identical to that specified by HL.
The $\overline{F_i}$ is are evaluated with Lindzen's (1971) WKB formalism for equatorial waves in shear. When
only infrared cooling acts to damp the waves the formulae for $\overline{F_i}$ are
$$\overline{F_0}(z) = A_0 \, exp\left( - \int_{17km}^{z} \frac{\alpha N}{k(c - \overline{u})^2} \, dz \right) \tag{2}$$

for the Kelvin wave, and
$$\overline{F_1}(z) = A_1 \, exp\left[ - \int_{17km}^{z} \frac{\alpha \beta N}{k^3 (c - \overline{u})^3} \left( 1 - \frac{k^2 (\overline{u} - c)}{\beta} \right) dz \right] \tag{3}$$

for the mixed Rossby-gravity wave. As in HL, the wavenumber $k$, the phase speed $c$, and $A_0$ are chosen
to be $2\pi/(40{,}000 \text{ km})$, $30 \text{ m s}^{-1}$, and $0.04 \text{ m}^2 \text{ s}^{-2} \rho_0(17 \text{ km})$, respectively for the Kelvin wave while
they are equal to $-2\pi/(10{,}000 \text{ km})$, $-30 \text{ m s}^{-1}$, and $-0.04 \text{ m}^2 \text{ s}^{-2} \rho_0(17 \text{ km})$, respectively for the
mixed Rossby-gravity wave. In Eq. (1), $K_z = 0.3 \text{ m}^2 \text{ s}^{-1}$, which is also the same as in HL. In addition,
$\beta = 2\Omega/a$, where $\Omega$ is earth's rotation rate, and $a$ is earth's radius. HL's boundary conditions stipulated
that $\overline{u} = 0$ at the lowest model level (17 km) and constrained $\overline{u}$ to vary semiannually at the top level (35
km).
In our control run that is used to depict the present-day QBO all the model parameters are identical
to those used by HL in their original simulation. The Brunt-Väisälä frequency
$$N = \sqrt{\frac{g}{T_0}\left(\frac{dT_0}{dz} + \frac{g}{c_p}\right)} \tag{4}$$

In Eq. (4), $g$ is gravity, $T_0$ is mean temperature, and $c_p$ is specific heat of dry air at constant pressure.
HL set $N$ in Eq. (4) to $2.16 \times 10^{-2} \text{s}^{-1}$ with a scale height $H = 6 \text{ km}$. In addition, the Newtonian
cooling profile in our control run, i.e., $\alpha(z)$ in Eqs. (2) and (3), is also identical to that in the original
HL model and depicted in FIG. 1a as the black line. Namely, $\alpha(z)$ in the control run increases from
(21 day)$^{-1}$ at 17 km to (7 day)$^{-1}$ at 30 km and is kept at (7 day)$^{-1}$ between 30 km and 35 km. Fels
(1985) explained why the magnitude of this radiative damping rate is suitable for simulating the QBO
on the basis of the scale-dependent effect of radiative damping (Fels, 1982). Hamilton (1981)
demonstrated that the proper choice of $\alpha(z)$ is crucial in simulating a realistic vertical structure of the
QBO.
Eq. (1) was integrated for 100 years using the forward-backward scheme (Matsuno, 1966). The
vertical resolution was 250 m and identical to that in HL. The time step was 12 hr, i.e., one half of used
in HL, because the 24-hr time step resulted in numerical instability in our integration.
FIG. 2a shows the time–height section of the monthly averaged mean zonal wind simulated over the
first 20 years using the HL model. Both the QBO and the semiannual oscillation (SAO) are conspicuous.
The fast Fourier transform (FFT) method is used to calculate the frequency power spectra. In order to
more accurately derive the QBO period, the model was run for 100 years to increase the spectral
resolution. Frequency–height sections of the power spectral densities (PSD) over zero to the Nyquist
frequency, i.e., 0.5 cycle/month, depict two sharp lines (peaks) at $\frac{1}{30}$ and $\frac{1}{6}$ cycle/month, respectively
(not shown). In order to better visualize the magnitudes of the PSD, we show two truncated frequency–
height sections with FIG. 2b and FIG. 2c highlighting the QBO and the SAO respectively. FIG. 2b shows
that the QBO dominates over the model domain. The peak frequency corresponds to the period of 30
months. FIG. 2c shows the SAO dominates near the model top due to the fact a semiannual forcing was
imposed in the altitudes from 28 to 35 km.
It is worth mentioning that the QBO period shown here is longer than 26.5 months reported in the HL
paper (see their FIG. 1). Using the HL model parameters, the QBO period simulated by Plumb (1977)
was close to three years (refer to his FIG. 8a), which is longer than our simulated QBO period, i.e., 30.0
months. Although we could not explain why our simulated QBO period is longer than that simulated by
HL, we found that when the upper boundary condition is changed from $\overline{u} = 14 \sin(\omega_a t)$ and $\omega_a =$
$\frac{2\pi}{180}$ day$^{-1}$ used in the HL's original model (refer to their Eqs. (2)) to $\frac{\partial \overline{u}}{\partial z} = 0$ used in Plumb (1977), the
simulated QBO period becomes 34.3 month (figure not shown). In other words, when we adopted the
stress-free upper boundary condition as in Plumb (1977), our simulated QBO period is comparable to
that simulated by him, which lends credence to our reconstruction of the HL model.

In order to rigorously quantify the relationship of the Newtonian cooling coefficient at any altitude $z$

between the reference and doubled $CO_2$, we follow Dickinson (1973) in using a radiative transfer model
to calculate $Q_1(T)$ for a reference temperature profile $T(z)$ and $Q_1(T + \delta)$ for $T(z) + \delta$, where a small
perturbation $\delta T = 0.1\ K$ with $T(z)$ being the 1976 U.S. standard atmosphere. Our radiative transfer
computations use the MODTRAN gas absorption database with 0.1 cm$^{-1}$ spectral resolution (Jin et al.
2019; Berk et al. 2008). We then repeat the computations with the doubled $CO_2$ to yield $Q_2(T)$ and
$Q_2(T + \delta)$. It follows that $\frac{\alpha_2}{\alpha_1} = \frac{Q_2(T+\delta) - Q_2(T)}{Q_1(T+\delta) - Q_1(T)}$, where $\alpha_1(z)$ and $\alpha_2(z)$ stand for the Newtonian cooling
coefficient at any altitude $z$ for the reference and doubled CO2, respectively. In FIG. 1b the black line
depicts the ratio for the broadband longwave radiation ($5\ \mu m - 100\ \mu m$) and the red line delineates the
ratio for the $CO_2$ absorption band ($12\ \mu m - 18\ \mu m$). The ratio calculated over the broadband is
conspicuously smaller than that for the $CO_2$ absorption band, because the changes in cooling rate from
the temperature perturbation are larger over a wider spectral band. It is worth mentioning that the ratios
calculated over the broadband in the middle stratosphere are close to 1.3 and comparable to what Fels
(1985) estimated, i.e., about 23% decrease of the radiative damping time under a doubling of $CO_2$.

Returning to the 1D HL model, we synthesize those findings by prescribing $\alpha_2(z)$ in our

experimental runs for the doubled $CO_2$ as follows: an increase of 30% between 30 and 35 km, no change
below 24 km, and linear interpolation between 24 and 30 km. The resulting increase of radiative damping
rate from the control runs is depicted as the red line in FIG. 1a. This increase is reasonable based on our
results shown in FIG. 1b.
FIG. 3a shows the time–height section of the monthly averaged mean zonal wind simulated over the
first 20 years for the doubled $CO_2$ run, where the increased $\alpha(z)$ depicted as the red line in FIG. 1a was
employed while all other parameters are identical to those in the control run. Obviously, the QBO
dominates below 28 km while the semiannual oscillation (SAO) dominates above 31 km. Like FIG. 2b
and FIG. 2c, we only show two truncated frequency–height sections with FIG. 3b highlighting the QBO
and FIG. 3c highlighting the SAO. FIG. 3b also shows that the QBO prevails over the model domain. The
peak frequency corresponds to the period of 27.9 months. FIG. 3c shows the SAO dominates near the
model top due to the same imposed semiannual forcing as that in the control run.
In summary, using the original HL model we found that the increased radiative damping due to the
doubling of CO2 shortens the QBO period by 7% (i.e., decreases from 30 months to 27.9 months).

**3. Sensitivity of the QBO period to enhanced stratospheric radiative damping in the modified HL**
**model without the semiannual forcing**
HL pointed out that the imposed semiannual oscillation was not essential for their QBO theory.
Applying $\frac{\partial \overline{u}}{\partial z} = 0$ as the upper boundary condition, Plumb (1977) showed a simulated QBO without
resorting to the semiannual momentum source (refer to his FIG. 8b). In the following control run, all
parameters are identical to those used in the previous control run in Section 2 except that $G$ in Eq. (1) is
set to zero with $\frac{\partial \overline{u}}{\partial z}$ also being set to zero at $z = 35$ km. Hereafter we refer to it as the Plumb model[1]. FIG.

---

[1] Strictly speaking, it is the HL model modified by Plumb (1977). In this paper, we don't use his eponymous model, i.e., the simplest possible model of the QBO, where Boussinesq fluids with uniform mean density were employed, because the HL model and its variant are considerably more realistic.

4a shows the time–height section of the monthly averaged mean zonal wind simulated over the first 20
years using the Plumb model. As expected, the QBO emerges without any trace of SAO since $G = 0$ in
Eq. (1). FIG. 4b shows that the QBO dominates over the whole model domain. The peak frequency
corresponds to the period of 37.5 months, which is comparable to that simulated by Plumb (1977) shown
in his FIG. 8b. Apparently, the QBO period from the Plumb model, i.e., 37.5 months shown in FIG. 4b,
is longer than that from the HL model, i.e., 30.0 months shown in FIG. 2b. This is partly because the
additional forcing $G$ in Eq. (1) was removed in the Plumb model.

In the following experimental run, all parameters are identical to those used in the previous

experimental run in Section 2 except that $G$ in Eq. (1) is set to zero with $\frac{\partial \overline{u}}{\partial z}$ also being set to zero at $z =$
35 km. In other words, the following experimental run using the Plumb model employed the same
parameters as the afore-mentioned control run using the Plumb model except that the increased $\alpha(z)$
shown as the red line in FIG. 1a was used in the following experimental run while $\alpha(z)$ shown as the
black line in FIG. 1a was used in the above control run. FIG. 5a shows the time–height section of the
monthly averaged mean zonal wind simulated over the first 20 years for the doubled $CO_2$ run. It is natural
that only the QBO emerges. A comparison of FIG. 4a and FIG. 5a shows that the QBO period shortens
when the infrared damping increases in response to the doubled CO2. FIG. 5b shows that the QBO
dominates over the whole model domain. The peak frequency corresponds to the period of 31.6 months.

Using the Plumb model, we found that the increased radiative damping due to the doubling of $CO_2$

shortens the QBO period by 15.7% (i.e., decreases from 37.5 months to 31.6 months).

**4. Discussion**

Dunkerton (1997) showed that in the presence of tropical upwelling it was gravity waves rather than

large-scale Kelvin and mixed Rossby-gravity waves that contributed the bulk of QBO forcing.
Consequently, Geller et al. (2016a, 2016b) pointed out that enough gravity wave momentum flux is
required to model the QBO in a self-consistent manner in climate models and that the magnitude of the
subgrid-scale gravity wave momentum flux plays a crucial role in determining the QBO period. Since
there is no tropical upwelling in either the HL model or the Plumb model, it is natural that planetary-
scale Kelvin and mixed Rossby-gravity waves largely determine the QBO periods shown in Sections 2
and 3 due to the fact that the specified $G$ is significantly weaker than that in the terrestrial stratosphere.
We conducted another sensitivity test where all parameters are identical to those in the HL model except
that $G$ in both the control and experimental runs is twice as large as that used by HL. As the radiative
damping profile changes from the black line to the red line above 24 km shown in FIG. 1a, our simulated
QBO period decreases from 28.6 months to 27.3 months (figures not shown). This smaller percentage
decrease of 4.5% is not unexpected because $G$ is not sensitive to the radiative damping at all and the
greater specified $G$ reduces the fraction of the total wave forcing arising from the planetary waves.

We further conducted two sensitivity tests where all parameters are identical to those in the HL model

except that $G$ in the first test is half as large as that used by HL and is equal to zero in the second test.
Surprisingly, as the radiative damping profile changes from the black line to the red line above 24 km
shown in FIG. 1a, our simulated QBO periods decreases from 30.0 months to 28.6 months both for $G$
being decreased by 50% and for $G = 0$ (figures not shown). This 4.7% decrease in the QBO period is
smaller than the 7% reduction obtained from the sensitivity test presented in Section 2 when $G$ is the
same as that used by HL. It is surprising because the model atmosphere is expected to be more sensitive
to the changes in the radiative damping as $G$ becomes smaller and smaller. Note that when our control
runs adopt the black radiative damping profile shown in FIG. 1a the simulated QBO periods are not
sensitive to the imposed semiannual forcing provided that $G$ does not exceed the values employed by
HL. Similarly, when our experimental runs adopt the red radiative damping profile above 24 km shown
in FIG. 1a the simulated QBO periods are also not sensitive to the imposed semiannual forcing provided
that $G$ does not exceed 50% of the values adopted in HL. The question naturally arises: what is
responsible for this unexpected behavior?

In Section 2, the simulated QBO periods are equal to 30 and 34.3 months when we adopted the no-

slip and stress-free upper boundary condition respectively with all other parameters being identical to
those used by HL. The results implicate the upper boundary conditions in the inconsistency. Plumb (1977)
pointed out that the upper boundary in HL was undesirably low and implied that raising the lid to an
additional 50% would be adequate for the robustness in his model. Here, we carry out a series of
sensitivity tests by raising the model lid gradually from 35 km to 55 km with the one-kilometer increment.
we will demonstrate how the behavior of the HL model with $G = 0$ converges with that of the Plumb
model. The modified HL model, i.e., the HL model with $G = 0$ is identical to the Plumb model except
that the former has the no-slip upper boundary condition while the latter has the stress-free upper
boundary condition. Both models share the same governing equation (5). Note that we set the radiative
damping rate above the 35 km level to its value at the 35 km level shown in FIG. 1a.

For the radiative damping profile corresponding to the reference $CO_2$, FIG. 6 shows that when the

model lid is placed at the 35 km level the simulated QBO period of 30.0 months with the no-slip upper
boundary condition (solid black line) is apparently shorter than that of 37.5 months with the stress-free
upper boundary condition (dashed black line). FIG. 6 also shows that as the model lid is raised
incrementally from the 35 km level to the 46 km level, the discrepancies between the simulated QBO
periods due to the different upper boundary conditions decrease monotonically. No matter whether we
adopt the no-slip or stress-free upper boundary condition, the simulated QBO period is 32.4 months for
the reference radiative damping profile provided that the model top is at or above the 46 km level.
Similarly, for the radiative damping profile corresponding to the doubled $CO_2$, FIG. 6 demonstrates
that when the model lid is placed at the 35 km level the simulated QBO period of 28.6 months with the
no-slip upper boundary condition (solid red line) is obviously shorter than that of 31.6 months with the
stress-free upper boundary condition (dashed red line). FIG. 6 also exhibits that as the model lid is raised
gradually from the 35 km level to the 40 km level, the discrepancies between the simulated QBO periods
due to the different upper boundary conditions decrease monotonically. No matter whether we adopt the
no-slip or stress-free upper boundary condition, the simulated QBO period for the enhanced infrared
cooling due to the doubled $CO_2$ is 30.0 months provided that the model top is at or above the 40 km level.
It is apparent that the required model top is lower when the radiative damping is augmented due to the
doubling of $CO_2$ because the planetary waves dissipate more steeply with height in presence of the
enhanced infrared cooling rates.
FIG. 6 suggests that when the model lid is sufficiently high the QBO period in response to the
enhanced radiative damping due to the increasing $CO_2$ will decrease from 32.4 to 30.0 months. This
7.4% decrease in the QBO period is independent of the upper boundary condition. Note that the relative
uncertainty in the ratio $\frac{\alpha_2}{\alpha_1}$ calculated over the broadband (refer to the black line shown in FIG. 1b) ranges
from 5% to 10% in the lower stratosphere and from 10 to 15% in the middle and upper stratosphere.
Thus, the relative uncertainty in the calculated ratio is 15% at a liberal estimate in the stratosphere. Using
the HL model with its top at the 48 km level, we further conducted two experiments by adopting $G = 0$
in Eq. (1) and increasing the radiative damping corresponding to the doubled $CO_2$ between 30 km and
48 km by $30\% - 30\% * 15\% = 25.5\%$ and $30\% + 30\% * 15\% = 34.5\%$ respectively relative to that
in the control run. The simulated QBO periods are 30.3 and 29.7 months respectively. Therefore, when
the model lid is sufficiently high the QBO period in response to the enhanced radiative damping due to
the doubled $CO_2$ will decrease by approximately $7.4\% \pm 0.9\%$.
Jonsson et al. (2004) showed that the doubled $CO_2$ induces a substantial cooling throughout most of
the middle atmosphere, which in turn increases the ozone mixing ratio by 15–20% in the upper
stratosphere and by 10–15% in the lower mesosphere (refer to their Figure 6). Incorporating this increase
into the ozone profile for the doubled $CO_2$, we recalculated the ratio of $\alpha_2$, the Newtonian cooling
coefficient for the doubled $CO_2$, to $\alpha_1$, the Newtonian cooling coefficient for the reference $CO_2$. Our
calculated $\alpha_2/\alpha_1$ is only slightly increased as compared with that shown FIG. 1b no matter whether the
$CO_2$ absorption band is $5\ \mu m - 100\ \mu m$ or $12\ \mu m - 18\ \mu m$ (figure not shown). It is not unexpected
because the infrared radiative cooling by ozone is significantly smaller than that by $CO_2$ (refer to Fig. 1
in Dickinson 1973) and, as a result, the 15–20% increases in the ozone mixing ratio will not make a
noticeable difference. Since the monthly and zonal mean 2-D ozone concentrations are specified in about
80% of CMIP5 models (Cionni et al., 2011) and the monthly mean 3-D ozone data are employed in
many CMIP6 models (Keeble et al., 2020),  it is expected that the change in the radiative damping due
to the increase in ozone in response to the doubled $CO_2$ only marginally impacts the QBO periods in
those models that do not include an interactive chemistry. However, the 15–20% increases in the ozone
mixing ratio in response to the doubled $CO_2$ do contribute the shortening of the QBO period due to its
role in strengthening the tropical upwelling in the stratosphere (Bushell et al., 2010).
The real atmosphere involves the complex interactions among dynamics, chemistry, and radiation
(Andrews et al., 1987). First of all, the dynamical QBO goes hand in hand with the ozone QBO (Hasebe,
1994). Shibata and Deushi (2005) pointed out that the radiative heating related to the ozone QBO could
modify the secondary meridional circulation associated with the QBO, and consequently could modify
how the QBO westerlies move down faster than the QBO easterlies, leading to the elongation of the
QBO period in the chemically interactive models as compared with the chemically non-interactive
models. However, it is difficult to fathom how the intensity of the secondary meridional circulation
associated with the QBO could change the QBO period due to the fact that any speedup or slowdown of
the descending westerly shear zones is roughly compensated by the concurrent slowdown or speedup of
the descending easterly shear zones. Furthermore, due to the increase of the zonal mean temperatures
and ozone concentrations with the altitude in the lower stratosphere, the motion, ozone, and thermal
waves are closely coupled. Pawson et al. (1992) found that the net linearized cooling coefficient from
$CO_2$ (Newtonian cooling) was more than compensated in the lower equatorial atmosphere arising from
absorption by the 9.6 $\mu m$ bands of ozone (see their figure 8). This reduced or negative radiative damping
in the lower stratosphere acts to lengthen the QBO period in presence of the increased ozone due to the
doubling of the $CO_2$ concentrations. Finally, taking into account the short-wave heating of eddy ozone,
Cordero et al. (1998) used a mechanistic model with a one-dimensional representation for mean flow
and a three-dimensional depiction for Kelvin and Rossby-gravity waves to demonstrate that the ozone
distribution, which maximizes in the middle stratosphere, leads to the local radiative damping decreases
by up to 15% below 35 km and its increases by up to 20%. They concluded that ozone feedbacks
lengthen the QBO period by about 2 months. Cordero and Nathan (2000) further developed a more
sophisticated mechanistic model with a two-dimensional representation for mean flow and a three-
dimensional depiction for Kelvin and Rossby-gravity waves, and surprisingly found that ozone
feedbacks had little influence on the QBO period. We believe that more studies should be conducted to
fully understand how to put together those various effects of interactive ozone on the QBO period.
Using the NASA Goddard Institute for Space Studies (GISS) Model E2.2-AP (Rind et al. 2020; Orbe
et al. 2020), DallaSanta et al. (2021), to the great extent, isolated the overall effect on the QBO period
of the increase in ozone due to the doubled $CO_2$. They first used a chemically non-interactive (NINT)
model to conduct two CMIP6 experiments: the preindustrial (pi) control run, and the doubled $CO_2$ (2X)
run. The two experiments only differ in the $CO_2$ concentration with the latter being two times the former.
Any other specification of these two experiments is the same, e.g., the ozone concentrations in the two
runs are identical. Their Figure 5 shows that the doubling of $CO_2$ in the NINT model shortens the QBO
period from 29.1 to 25 months. In other words, the doubling of $CO_2$ shortens the QBO period by 14%.
DallaSanta et al. (2021) then employed a chemically interactive CMIP6 model with a mass-based scheme
(Bauer et al., 2020), called One-Moment Aerosol (OMA), to conduct the pi control run and the 2X run.
The ozone concentrations simulated by the OMA model increase by 10-15% in response to the doubling
of $CO_2$ (figure not shown), which is consistent with the results of Jonsson et al. (2004). Figure 5 in
DallaSanta et al. (2021) indicates that the doubling of $CO_2$ in the OMA model shortens the QBO period
from 31.7 to 26.6 months. Namely, the doubling of $CO_2$ shortens the QBO period by 16%. In short, the
results of DallaSanta et al. (2021) appear to suggest that three-way interactions among dynamics,
chemistry, and radiation tend to slightly amplify the shortening of the QBO period in response to the
doubling of $CO_2$.
Note that $N$, the Brunt-Väisälä frequency, in Eqs. (2) and (3) also changes with increasing $CO_2$.
Richter et al. (2020b) showed that $N^2$ would be decreased by ~5% in the stratosphere when $CO_2$ is
doubled (refer to their Figure 2c). We used the HL model to conduct a sensitivity test by adopting $G = 0$
in Eq. (1) with the radiative damping profile corresponding to the doubled $CO_2$ and the top of the models
at the 48 km level. The rest of parameters in this sensitivity test are identical to those in all the previous
runs except that the Brunt-Väisälä frequency in this experimental run was 2.5% smaller than that in the
control run. The models were run for 1000 years to further increase the spectral resolution. We found
that when the Brunt-Väisälä frequency was decreased by 2.5%, the simulated QBO period was slightly
lengthened from 30 months to 30.2 months (figure not shown). In other words, the impact of decreasing
stratospheric buoyancy frequency on the QBO period is almost negligible.
Analyzing eleven CCMI-1 REF-C2 climate–chemistry simulations, Eichinger and Šácha (2020)
showed that the scale height in the stratosphere decreases by 2.3% per century. Accordingly, we used
the HL model to conduct another sensitivity test by adopting $G = 0$ in Eq. (1) with the radiative damping
profile corresponding to the doubled $CO_2$ and the top of the models at the 48 km level. The rest of
parameters in this sensitivity test are identical to those in all the previous control runs except that the
scale height in this experimental run was  2.3% smaller than that in the control run. The model was also
run for 1000 years for the sake of higher spectral resolution. We found that when the scale height was
decreased by 2.3%, the simulated QBO period was also shortened by about 2.3%, i.e., from 30 months
to 29.3  months (figure not shown). Apparently, the shortening of the QBO period due to the warming
climate is ascribed less to the shrinkage of the scale height in the stratosphere than to the enhancing of
the stratospheric radiative damping. Together, the shrinking scale height and the increasing radiative
damping shorten the QBO period by about 9.6%.

**5. Conclusions**
Plumb (1977) envisioned that stratospheric climate change would give rise to long-term changes in
the QBO period due to changes in radiative damping and the Brunt-Väisälä frequency. Using one-
dimensional (1D) models and taking into account the uncertainty due to the radiative damping rate, we
found that the enhanced radiative damping arising from the doubling of $CO_2$ leads to the shortening of
the QBO period by about 7.4% $\pm$ 0.9% provided that the model top is higher than the 46 km level.
Furthermore, when we incorporated both the 2.3% shrinkage of the scale height and the enhanced
radiative damping, the QBO period is shortened by about 9.6%. In addition, the impact of decreasing
stratospheric buoyancy frequency is marginal. While the increased ozone in response to the doubling of
$CO_2$ appears to  slightly further shorten the QBO period, more research needs to be done for the
appreciation of the underlying mechanisms. Note that our 1D models include neither gravity waves nor
tropical upwelling and assume that there are no changes in wave fluxes entering the equatorial
stratosphere.
From a comprehensive model perspective, Richter et al. (2020b) showed that the changes in period
of the QBO in warming climate simulations varied quite significantly among these models. Some models
projected longer mean periods and some shorter mean periods for the QBO in a future warmer climate.
They argue that uncertainty in the representation of the parameterized gravity waves is the most likely
cause of the spread among the QBOi models in the QBO's response to climate change.
In addition, CO2 increases in the NASA GISS Model E2.2-AP lead to a decrease of both QBO period
and QBO amplitude (DallaSanta et al., 2021). The period decrease is mostly associated with increases
in lower stratospheric momentum fluxes (related to parameterized convection), a finding consistent with
Geller et al. (2016a, 2016b) and Richter et al. (2020b). The amplitude decrease is mainly associated with
a strengthened residual mean circulation, also consistent with the literature, although the vertical
structure of the circulation response is nontrivial. It is worth mentioning that horizontal momentum flux
divergences could also play an important role in weakening the QBO (Match and Fueglistaler, 2019,

2020).

Our 1D models only explored how the QBO period responds to the enhancing radiative damping of
planetary waves, the shrinking scale height in the stratosphere, and the decreasing stratospheric
buoyancy frequency due to the increasing $CO_2$ concentration. In order to investigate how those factors
affect gravity waves which play an even more important role in determining the QBO period than
planetary waves, high-resolution models such as those used by Kawatani et al. (2011, 2019) are desirable
to further our understanding. Ultimately, how the QBO period changes in response to the increasing $CO_2$
will be determined by the combined effects of the strengthening of tropical upwelling, the increasing of
wave fluxes entering the equatorial stratosphere, the enhancing of radiative damping, and the shrinking
of the scale height in the stratosphere, which warrants further research.

**Data availability**
Any data used in this paper can be made available from the corresponding author upon request.

**Author contributions**
All authors made equal contributions to this work.

**Competing interests**
The authors declare that they have no conflict of interest.

**Acknowledgements:** Climate modeling at GISS is supported by the NASA Modeling, Analysis and
Prediction program, and resources supporting this work were provided by the NASA High-End
Computing (HEC) Program through the NASA Center for Climate Simulation (NCCS) at Goddard Space
Flight Center. KD acknowledges support from the NASA Postdoctoral Program. The authors thank the
editor Peter Haynes and two anonymous reviewers for their helpful comments, which led to an improved
paper. The authors also acknowledge very useful discussions with Drs. Geller and Orbe.

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

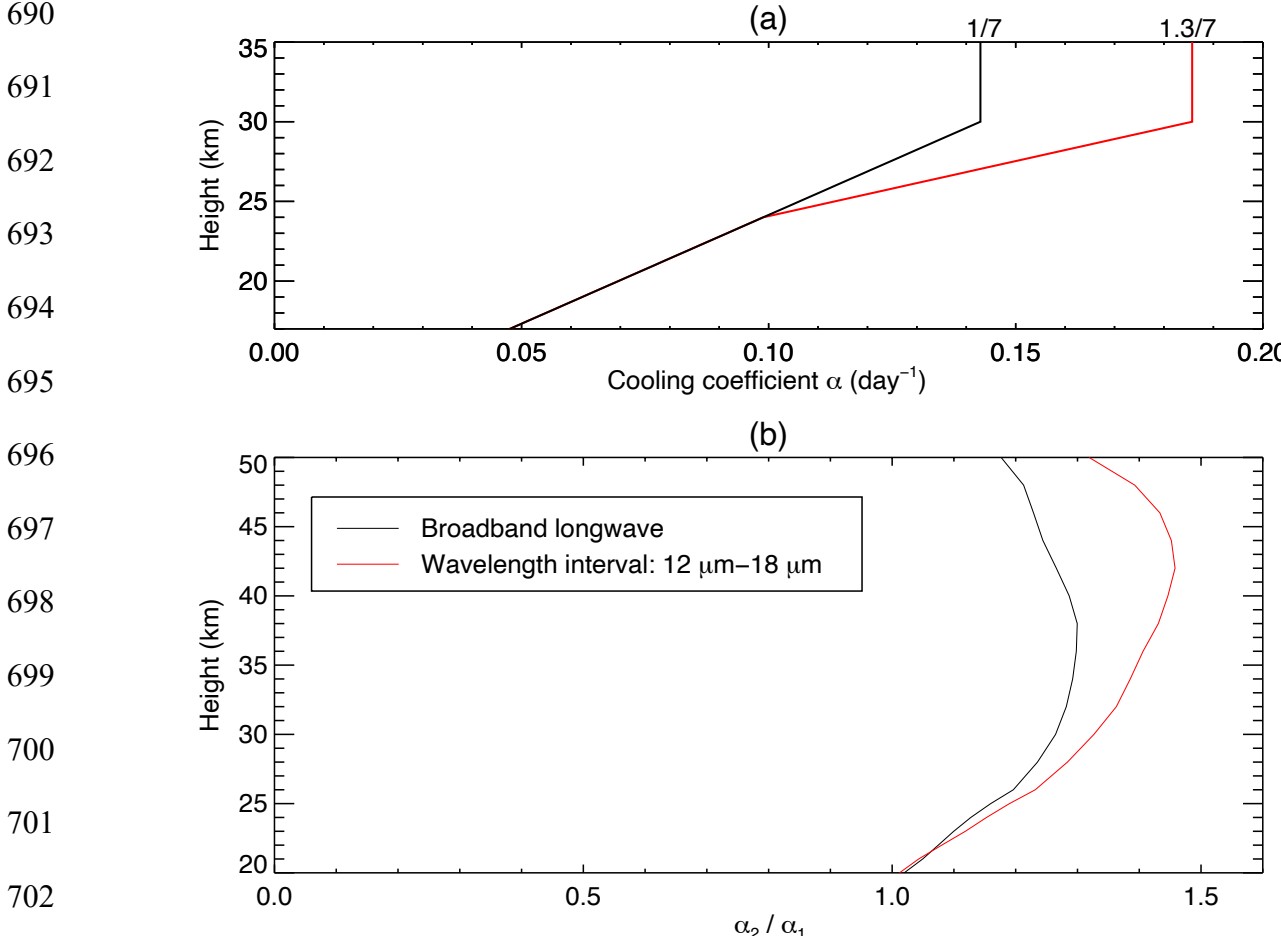

FIG. 1: (a) Profiles of Newtonian cooling coefficients: The smaller values (black line) are used for the control runs while the larger values (red line) are used for the experimental runs. (b) Profiles of the ratio of $\alpha_2$ to $\alpha_1$, where $\alpha_1$ and $\alpha_2$ denote the Newtonian cooling coefficient for the reference $CO_2$ and the doubled $CO_2$, respectively. The black line depicts the ratio for the broadband longwave ($5\ \mu m - 100\ \mu m$) and the red line delineates that for the $CO_2$ absorption band ($12\ \mu m - 18\ \mu m$).


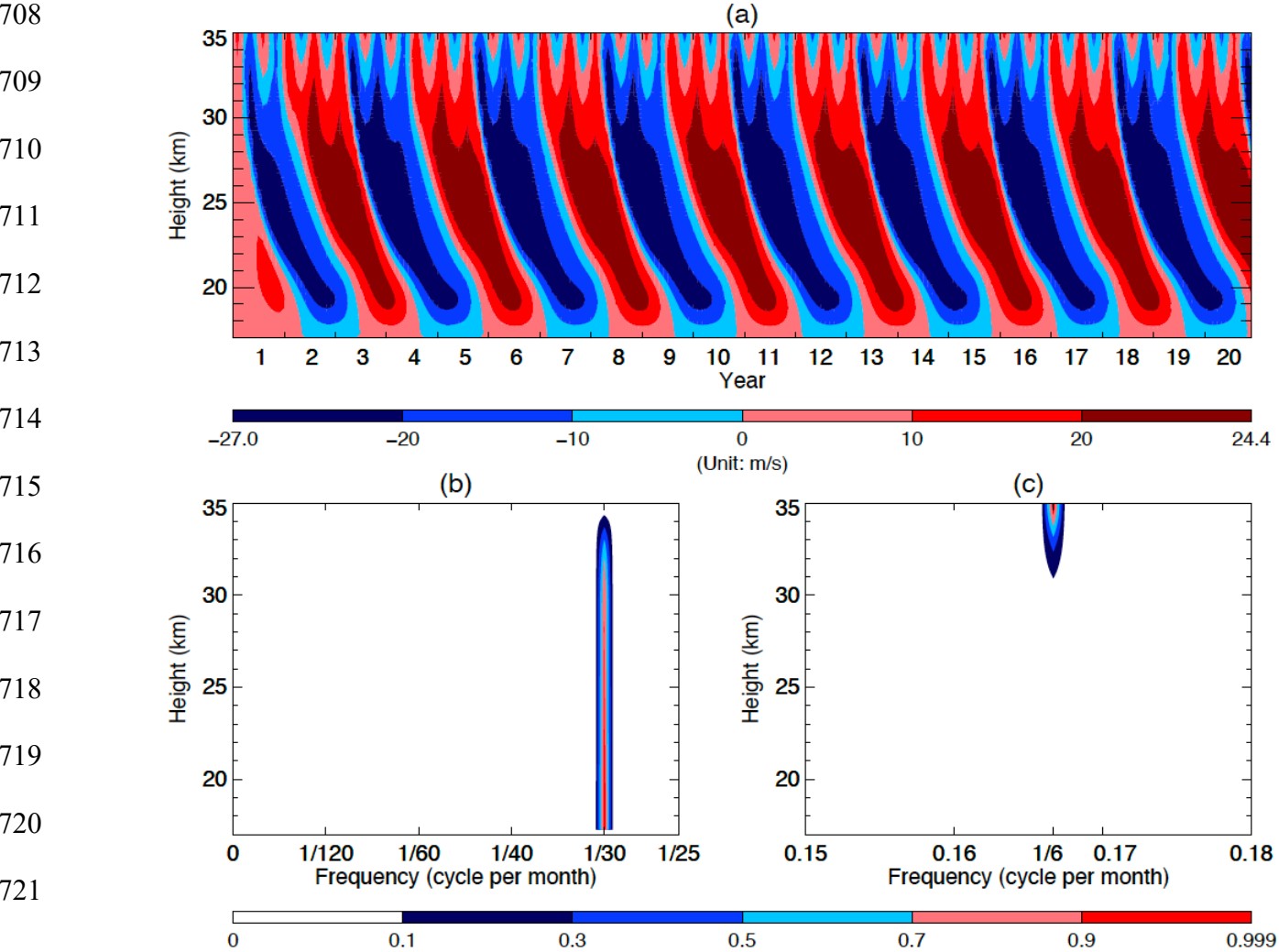

FIG. 2: (a) Time–height section of the monthly averaged mean zonal wind over the first 20 years from
the HL's original model. (b) and (c) Frequency–height section of the power spectral densities (PSD) of
the standardized monthly averaged mean zonal wind of the 100 years. Note that in order to better
visualize the PSD in (b) and (c), we trimmed off the blank segments for the frequencies ranging from
$\frac{1}{25}$ to 0.15 cycle per month and those ranging from 0.18 to 0.5 cycle per month.

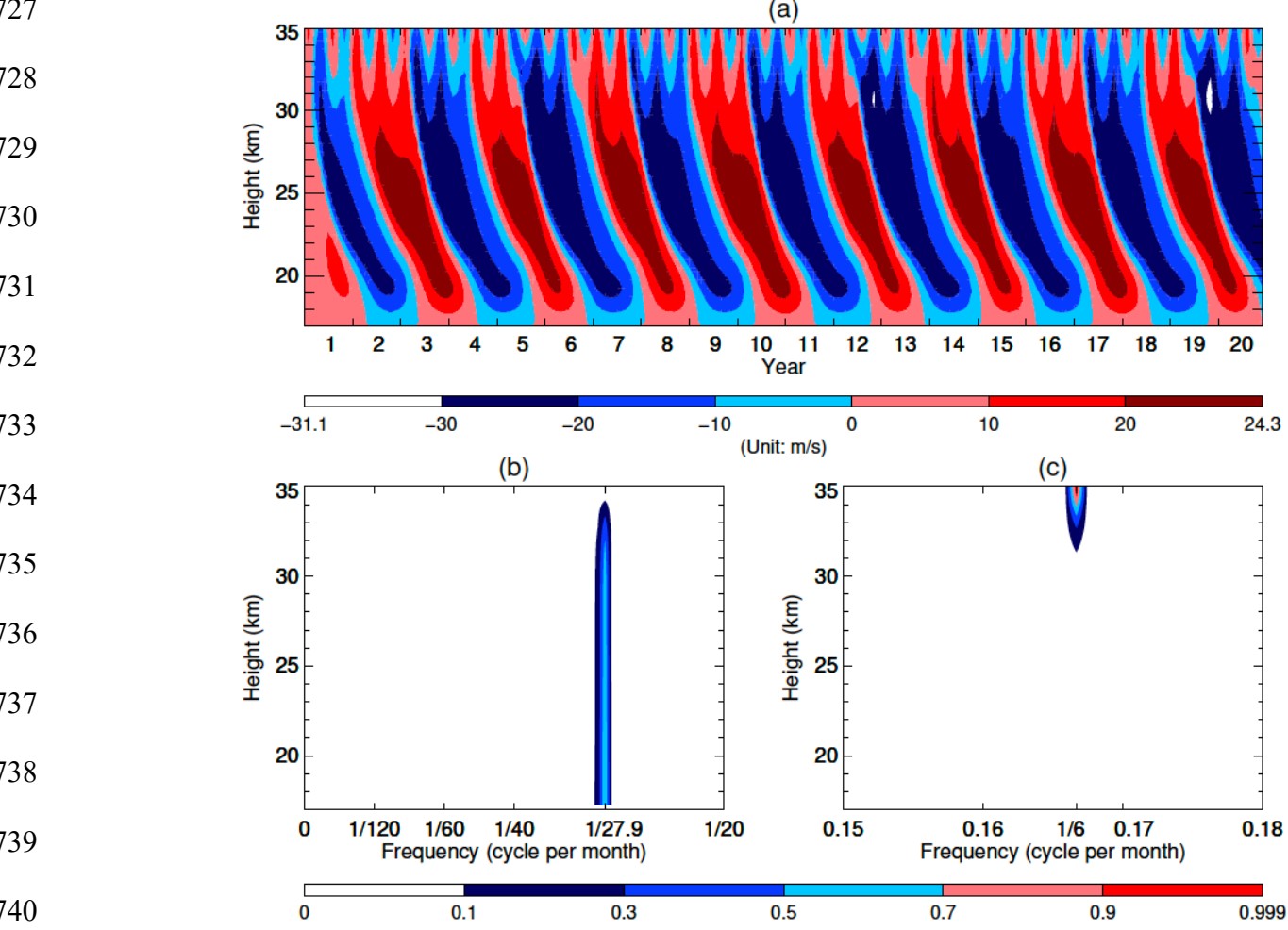

**FIG. 3**: (a) Same as FIG. 2a, but with the enhanced $\alpha(z)$ depicted as the red line in FIG. 1a. (b) and (c) Frequency–height section of the power spectral densities (PSD) of the standardized monthly averaged mean zonal wind of the 100 years. Note that in order to better visualize the PSD in (b) and (c), we trimmed off the blank segments for the frequencies ranging from $\frac{1}{20}$ to 0.15 cycle per month and those ranging from 0.18 to 0.5 cycle per month.

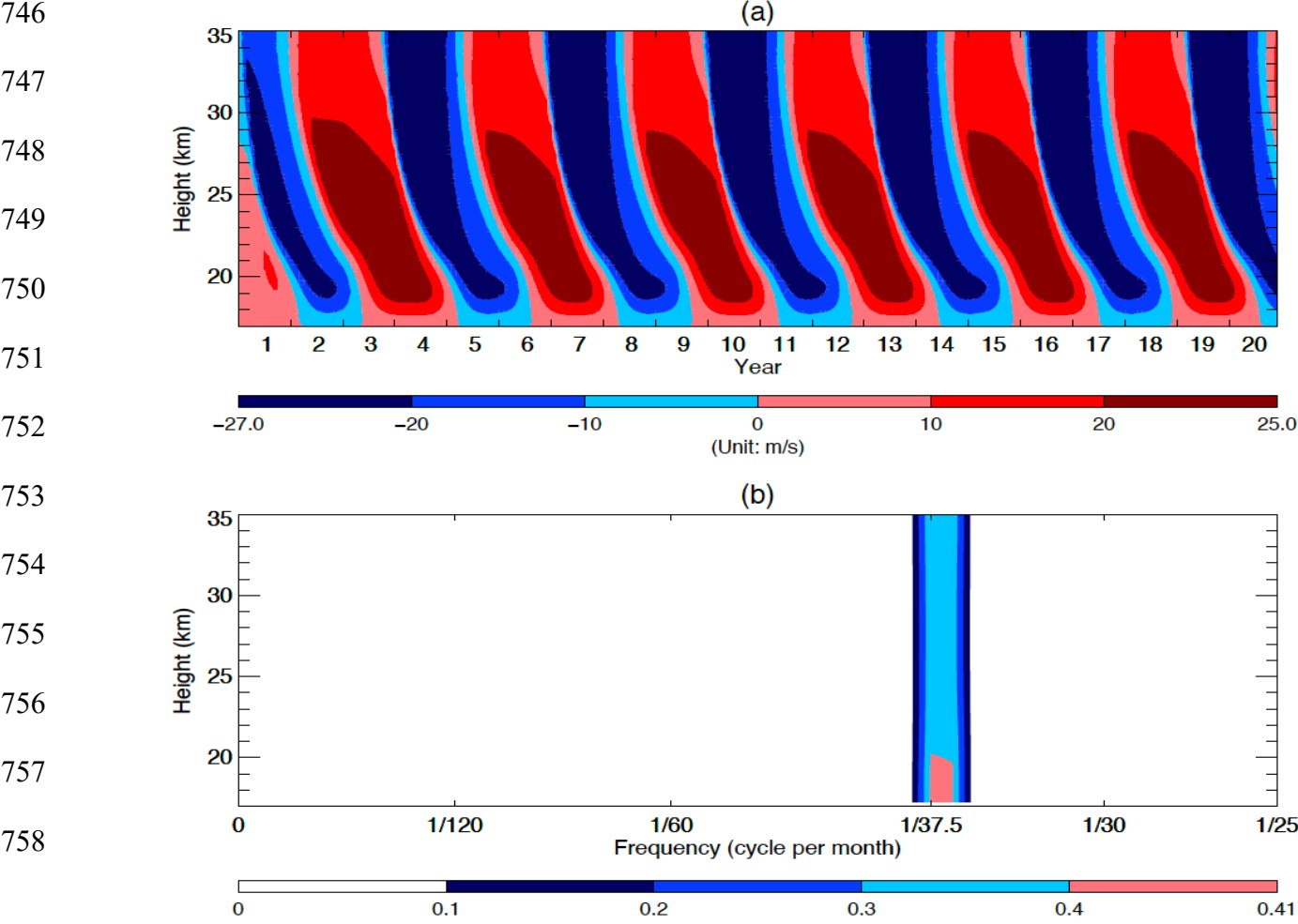

FIG. 4: (a) Time–height section of the monthly averaged mean zonal wind over the first 20 years from the HL's model without the semiannual forcing. (b) Frequency–height section of the power spectral densities (PSD) of the standardized monthly averaged mean zonal wind of the 100 years. Note that in order to visualize the PSD, we trimmed off the blank segment for the frequencies ranging from $\frac{1}{25}$ to 0.5 cycle per month.

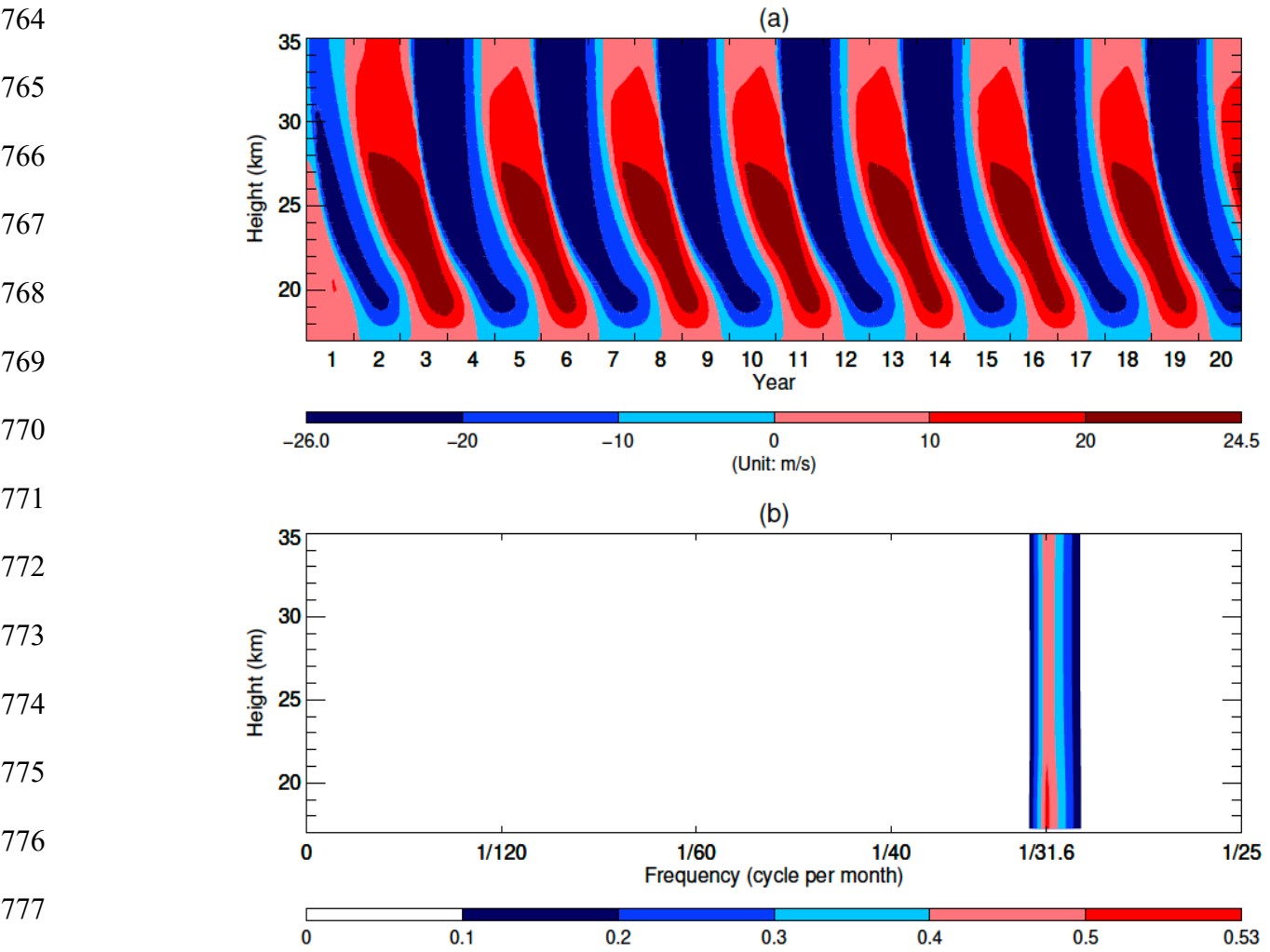

**FIG. 5**: (a) Same as FIG. 4a, but with the enhanced $\alpha(z)$ depicted as the red line in FIG. 1a. (b) Same as FIG. 4b, but for the doubled $CO_2$ Run.

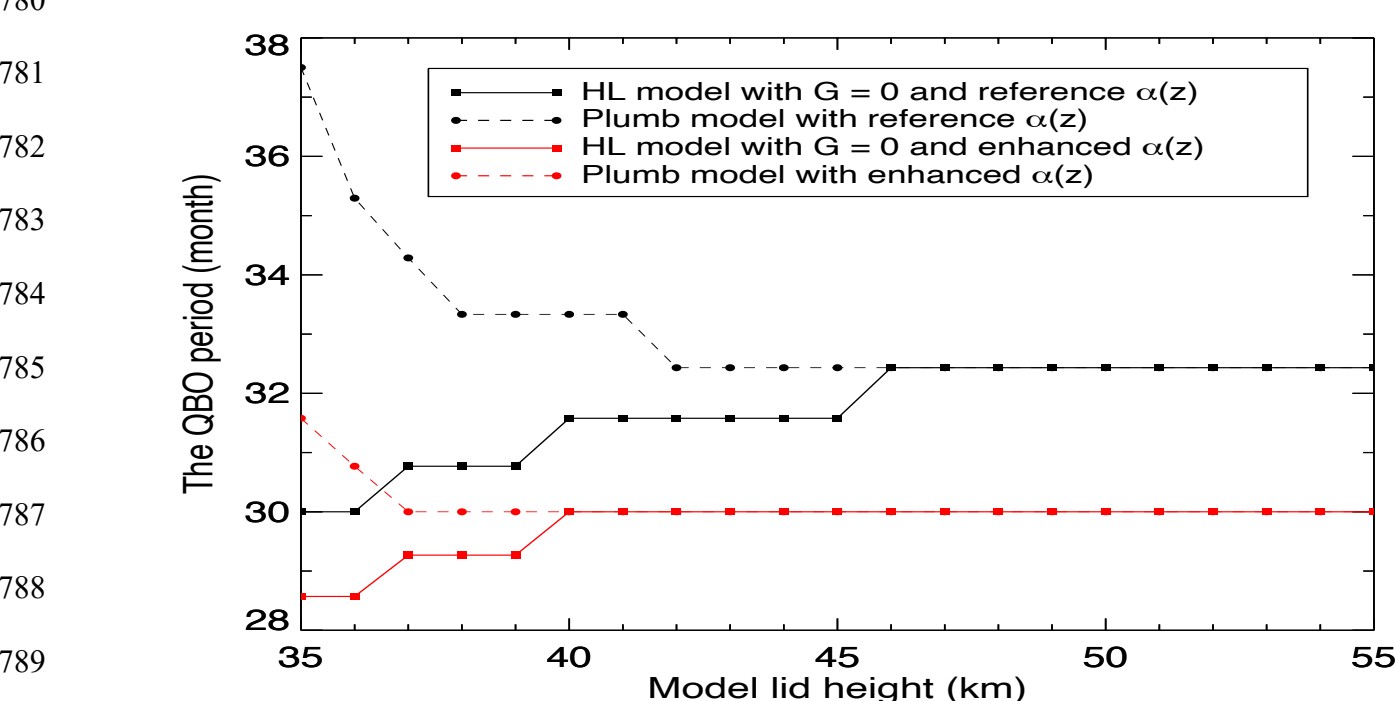

**FIG. 6**: The relationship between the simulated QBO period with the height of the model lid. Black and red lines depict the results from using the reference radiative damping and the enhanced radiative damping respectively while solid and dashed lines delineate the results from the HL model with $G = 0$ and the Plumb model respectively.