# Peer review of "The Impact of Increasing Stratospheric Radiative Damping on the QBO Period Tiehan Zhou1,2, Kevin DallaSanta1,3, Larissa Nazarenko1,2, Gavin A. Schmidt1, Zhonghai Jin1 2 1NASA Goddard Institute for Space Studies, New York, NY"

_Atmospheric Chemistry and Physics, 2020_

## Author Comment (AC1) · 7 Oct 2020

The following three references were left out in the preprint:

(1) Butchart, N.: The Brewer-Dobson circulation, Rev. Geophys., 52, 157–184, https://doi.org/10.1002/2013RG000448, 2014.

(2) Dunkerton, T. J.: The role of gravity waves in the quasi-biennial oscillation, J. Geophys. Res., 102, 26053–26076,http://dx.doi.org/10.1029/96JD02999, 1997.

(3) Geller, M. A., Zhou, T., and Yuan, W.: The QBO, gravity waves forced by tropical convection, and ENSO, J. Geophys. Res. Atmos., 121, 8886-8895, https://doi.org/10.1002/2015JD024125, 2016b.

[Figure]

The corrected version is submitted as a supplement.

Please also note the supplement to this comment:
https://acp.copernicus.org/preprints/acp-2020-925/acp-2020-925-AC1-supplement.pdf
* * *
[Figure]

**Supplement:**

[revised manuscript text omitted]

---

## Referee Comment (RC1) · Anonymous Referee #1 · 22 Oct 2020

**1  Overview**

**Suggestion: Major revisions**

This manuscript examines the impact of increasing stratospheric radiative damping on the period of the QBO. The scale of the increase in stratospheric radiative damping is based on results from a radiative transfer model in Plass (1956). Radiative damping causes dissipation of vertically propagating waves, which can lead to mean flow accelerations and internal oscillations, namely the Quasi-Biennial Oscillation (QBO) of the tropical stratosphere. This paper investigates the sensitivity of the QBO to increased radiative damping rate in a classical one-dimensional model of the QBO. It is reported that increased radiative damping would decrease the height scale of wave

dissipation, and would be expected to lead to modest decreases in the QBO period (by 5-15% depending on the model formulation). Comprehensive climate models do not produce robust projections of the future QBO period, disagreeing on the sign of any future change. This disagreement is primarily thought to arise from competition between increasing wave stress (which tends to reduce the period) and increasing up-welling (which tends to increase the period). The mechanism proposed in this paper is an additional process that could potentially impact the QBO period in the future, and could already be happening in reality and in comprehensive climate simulations of the QBO.

The identification and characterization of a new process that could lead to changes in the QBO period is a worthwhile endeavor, and is appropriate for publication in this journal. One-dimensional models of the QBO are appropriate tools for characterizing the existence, sign, and order of magnitude of this radiative-dynamical sensitivity. This paper is careful to show how the results are sensitive to the formulation of the model, and those sensitivities help contextualize the argument.

This paper has good potential, although at present the approach requires more justi-fication, and the presentation could benefit from easing some of the tension between competing objectives of interpretability and predictive value. First, the radiative damp-ing projections cited in the paper are of questionable relevance to the work presented. Second, the focus in the manuscript on producing a deterministic prediction of future period change appears to be inconsistent with the uncertainty stemming from the for-mulation of the model. If the radiative damping can be grounded on a more reliable basis, and the emphasis in the paper can be shifted to focus on the interpretation of the hypothesized mechanism and its attendant uncertainties without asking too much of its predictive value, then the paper can be recommended for publication. As such, major revisions are recommended.

**2 Major revisions**

To elaborate on the recommendations for major revision:

First, the manuscript relies on a projection of radiative damping from Plass (1956), who diagnosed radiative cooling rates with a fixed temperature profile in response to a doubling of CO2. However, the connection between the Plass analysis and the radiative damping rate is not obvious. Radiative cooling rate has units of $[K \ s^{-1}]$, whereas radiative damping rate has units of $[s^{-1}]$. The cooling rate results of Plass cannot be used to isolate a change in radiative damping rate because the Plass result also includes changes in radiative equilibrium temperature, both of which impact the radiative cooling. The usage of the Plass value of 50% should be either (1) justified in light of these considerations or (2) an alternative reliable estimate should be provided of the radiative damping rate response to CO2 doubling (an order of magnitude estimate is fine). If no projection of radiative damping rate with CO2 doubling exists in the literature, then one should be produced (e.g. using a radiative transfer model). Such a projection of radiative damping rate, necessary for the arguments in the paper, would constitute a valuable contribution in its own right.

Second, there is tension in the manuscript between the interpretability and predictive value of the results. Using the 1D model makes a strong decision in favor of interpretability, which is well justified by the approach and the results. Noting that the QBO period in the basic state and the response to changes in radiative damping are both highly sensitive to minor changes in the model formulation, it appears that the 1D model can provide, at best, the sign and order of magnitude of the period change in response to an increase in radiative damping. Therefore, modeling decisions that sacrifice the interpretability of the final results without impacting the sign or order of magnitude of the final result should be justified or avoided. Two example decisions in the paper are as follows. For each, the sacrifice of interpretability should be (1) justified in terms of predictive value or some other objective or (2) the simpler case should be considered:
- The Holton (1972) formulation used in this paper is driven by asymmetric waves (a Kelvin wave and a Rossby wave with different dispersion relations and wavenumbers). The Plumb (1977) formulation is driven by symmetric wave stress (equal and opposite gravity waves), allowing for clear interpretation of the model dynamics in terms of a small number of dimensionless parameters. Is there a benefit to using asymmetric wave forcing in this paper that justifies it at the cost of sacrificing the interpretability of the symmetric formulation?

- Another source of the tension is in the choice to include changes in the buoyancy frequency $N$ in the projections of the model response to radiative damping changes. Including the small (2.5%) changes in $N$ seems to be so marginal that its effects are primarily to sacrifice interpretability without a clear benefit. It can still be useful to include a sensitivity study to changes in $N$, but this sensitivity study should be distinguished from the main line of argumentation in the paper. Note that in the 1D model, the buoyancy frequency $N$ and radiative damping $\mu$ are always multiplied together, such that their combined effects are tantamount to considering a $(1.5 * 1.025 \rightarrow) \approx 54\%$ change in radiative damping (or buoyancy frequency) alone, not significantly different than the 50% change in radiative damping.

**3   Minor comments**

It would be appreciated if the following minor comments regarding the content and structure of the paper were addressed:

- The decomposition of $\bar{u} = \bar{u}_{SA} + \bar{u}_{QBO}$ in equation (5) gives the impression that the evolution of $\bar{u}_{QBO}$ depends only on $\bar{u}_{QBO}$, and that the influence of $\bar{u}_{SA}$ has been factored out. Yet $\bar{u}_{SA}$ still impacts the evolution of $\bar{u}_{QBO}$ through the wave

forcing $\bar{F}_i$. Because $\bar{F}_i$ is nonlinear in the zonal wind profile, the SAO will impact the wave forcing, which changes the mean wind and then alters the diffusion profile. So, the dynamics are not simply resulting from the sum of a linear QBO dynamic and a linear SAO dynamic. Given the limitations of this decomposition, what benefit is provided by its inclusion?

• Lines 301-312 list all numerical values from Figure 6. Is it necessary to list all numerical values when the figure provides their approximate value? If so, then perhaps a table can be supplied instead of the figure or in the supplementary information. I suspect that the relevant information from Figure 6 can be conveyed in a more concise way.

• In the Introduction, the following question is raised: "Does the competing effect between [upwelling and enhanced wave stress] leave some room for increasing stratospheric radiative damping to exert an influence on the QBO period?" (Line 137) In the Conclusions, comprehensive QBO models are noted to have significant variance in their projections of period. A quantitative comparison would be useful here; allowing that the 1D model provides at best an order of magnitude estimate, is there reason to believe that period changes in GCMs are small enough that radiative damping could potentially impact their sign? Their magnitude? Or on the contrary, are the period changes in GCMs large enough in magnitude that radiative damping would not be expected to have a significant bearing on the sign or magnitude of the change?

• Lines 343 - 345: Recently, doubt has been cast on the role of upwelling on QBO amplitude (Match and Fueglistaler, 2019, 2020). A more nuanced assessment would be appreciated than "The amplitude decrease is associated with a strengthened residual mean circulation, also consistent with the literature, although the vertical structure of the circulation response is nontrivial."

**4 Edits**

(166) "mixing Rossby-gravity wave" should be "mixed Rossby-gravity wave"

**5 References**

- Holton, J. R., and R. S. Lindzen, 1972: An Updated Theory for the Quasi-Biennial Cycle of the Tropical Stratosphere. Journal of the Atmospheric Sciences, 29 (6), 1076-1080

- Match, A., and S. Fueglistaler, 2019: The Buffer Zone of the Quasi-Biennial Oscillation. Journal of the Atmospheric Sciences, 76 (11), 3553-3567

- Match, A., and S. Fueglistaler, 2020: Mean flow damping forms the buffer zone of the Quasi-Biennial Oscillation: 1D theory. Journal of the Atmospheric Sciences, 77 (6): 1955-1967.

- Plass, G. N., 1956: The influence of the 15p carbon-dioxide band on the atmospheric infra-red cooling rate. Tech. rep. doi:10.1002/qj.49708235307.

- Plumb, R. A., 1977: The Interaction of Two Internal Waves with the Mean Flow: Implications for the Theory of the Quasi-Biennial Oscillation. Journal of the Atmospheric Sciences, 34 (12), 1847-1858

---

## Referee Comment (RC2) · Anonymous Referee #2 · 8 Nov 2020

This is the first review of the manuscript "The Impact of Increasing Stratospheric Radiative Damping on the QBO Period" by Tiehan Zhou, Kevin DallaSanta, Larissa Nazarenko, Gavin A. Schmidt, Paper # acp-2020-925. The approach of the paper is to conduct sensitivity experiments using 1-D mechanistic model to find an impact of radiative dumping in the stratosphere to QBO period. Experimental parameters in this paper are Newtonian cooling (alpha; unit:s-1)/Brunt-Vaisala frequency (N; unit: s-1), and the upper boundary conditions (G). Diagnostics are monthly zonal wind, and the frequency power spectra using the fast Fourier transform. The results are interesting and relevant to Atmospheric Chemistry and Physics because this paper is trying to ascertain the trend of the QBO period in a warming climate, focusing on the radiative damping that would influence that period. But I suggest a major substantial revision

since I have serious concerns about generalization and validation of authors' results. (mainly described in the line-by-line comments).

Major comments

1. Statistical significance L047 "the doubling of CO2 shortens the QBO period by 4.7%" What kind of meaning does a value of 4.7 hold for the science? Results of sensitivity experiments using 1D model mostly depend on an assumption of the experimental design, here in Newtonian cooling profiles of Fig. 1. Can you show realistic vertical profiles of Newtonian cooling with standard deviations? And then, you can estimate errors about the shortening of the QBO period, from an assumption of errors of Newtonian cooling.

2. Scale height Scale height would be changed in a warming climate. How does the change of scale height due to the temperature change in a warming climate affect the QBO period?

3. Ozone The ozone also affects QBO period. It is useful for readers to assess an effect of the ozone on the QBO period using 1D model in a warming climate. Shibata, K., and M. Deushi (2005), Radiative effect of ozone on the quasi-biennial oscillation in the equatorial stratosphere, Geophys. Res. Lett., 32, L24802, doi:10.1029/2005GL023433.

L10-16. Why the authors do not show that QBOs simulated in this paper are derived only from planetary waves and that gravity waves are not included? Without manifestation about exploring the response to enhancing radiative damping of only planetary waves in the experiments, their conclusion would lead to misleading.

L15-16. Most climate models project a strengthening of tropical vertical residual velocity, as you mentioned in the introduction. This could contribute to projecting a lengthening of the QBO period, which is opposite direction to the authors' conclusion.

L232. "QBO was not essential for QBO theory" You can estimate QBO power
with/without SAO. To what extent does the SAO impact the QBO power spectrum?

Minor comments L301-323. Redundant descriptions. You can omit most of them.

---

## Short Comment (SC1) · 11 Nov 2020

Dear authors,

thank you very much for an interesting study. In connection with the comments raised by the referees, I would like to point to you a paper that we recently published (Eichinger and Šácha, 2020) on the impact of scale height changes on tropical upwelling trends. Stratospheric cooling due to increasing GHGs is both observed and robustly simulated by CCMs and it is directly related with scale-height changes. In our paper you can find the rate of "shrinkage" among CCMs and how this small change may result in considerable overestimation of the upwelling trends.

Best regards, Petr Šácha.

[Figure]

Eichinger, R, Šácha, P. Overestimated acceleration of the advective Brewer–Dobson circulation due to stratospheric cooling. QJR Meteorol Soc. 2020; 1– 15. https://doi.org/10.1002/qj.3876

---

## Author Comment (AC2) · 19 Nov 2020

Dear Petr,

Based on your paper, an additional sensitivity test has been conducted to examine how the scale height shrinkage impacts the QBO period. We are adding it to the manuscript.
* * *

---

## Author Comment (AC3) · 4 Dec 2020

**Response to Reviewer 1**

**1 Overview**

**Suggestion: Major revisions**

This manuscript examines the impact of increasing stratospheric radiative damping on the period of the QBO. The scale of the increase in stratospheric radiative damping is based on results from a radiative transfer model in Plass (1956). Radiative damping causes dissipation of vertically propagating waves, which can lead to mean flow accelerations and internal oscillations, namely the Quasi-Biennial Oscillation (QBO) of the tropical stratosphere. This paper investigates the sensitivity of the QBO to increased radiative damping rate in a classical one-dimensional model of the QBO. It is reported that increased radiative damping would decrease the height scale of wave dissipation, and would be expected to lead to modest decreases in the QBO period (by 5-15% depending on the model formulation). Comprehensive climate models do not produce robust projections of the future QBO period, disagreeing on the sign of any future change. This disagreement is primarily thought to arise from competition between increasing wave stress (which tends to reduce the period) and increasing upwelling (which tends to increase the period). The mechanism proposed in this paper is an additional process that could potentially impact the QBO period in the future, and could already be happening in reality and in comprehensive climate simulations of the QBO.

The identification and characterization of a new process that could lead to changes in the QBO period is a worthwhile endeavor, and is appropriate for publication in this journal. One-dimensional models of the QBO are appropriate tools for characterizing the existence, sign, and order of magnitude of this radiative-dynamical sensitivity. This paper is careful to show how the results are sensitive to the formulation of the model, and those sensitivities help contextualize the argument. This paper has good potential, although at present the approach requires more justification, and the presentation could benefit from easing some of the tension between competing objectives of interpretability and predictive value. First, the radiative damping projections cited in the paper are of questionable relevance to the work presented. Second, the focus in the manuscript on producing a deterministic prediction of future period change appears to be inconsistent with the uncertainty stemming from the formulation of the model. If the radiative damping can be grounded on a more reliable basis, and the emphasis in the paper can be shifted to focus on the interpretation of the hypothesized mechanism and its attendant uncertainties without asking too much of its predictive value, then the paper can be recommended for publication. As such, major revisions are recommended.

We thank you for your insightful comments and suggestions and will address them point by point as below.

**2 Major revisions**

To elaborate on the recommendations for major revision:

First, the manuscript relies on a projection of radiative damping from Plass (1956), who diagnosed radiative cooling rates with a fixed temperature profile in response to a doubling of $CO_2$. However, the connection between the Plass analysis and the radiative damping rate is not obvious. Radiative cooling rate has units of $[K\ s^{-1}]$, whereas radiative damping rate has units of $[s^{-1}]$. The cooling rate results of Plass cannot be used to isolate a change in radiative damping rate because the Plass result also includes changes in radiative equilibrium temperature, both of which impact the radiative cooling.

In his last section (i.e., section 4), Plass (1956) mentioned "The change in the equilibrium temperature at the surface of the earth with $CO_2$ concentration…" in order to counter "The argument has sometimes been advanced that the $CO_2$ cannot cause a temperature change at the surface of the earth because the $CO_2$ band is always black at any reasonable concentration…"

When Plass (1956) calculated the radiative cooling rates in other sections, he didn't deal with radiative equilibrium temperature at all. Instead, he obtained them for $\frac{1}{2} \times CO_2$, $1 \times CO_2$, and $2 \times CO_2$ with a fixed temperature profile.

Let's quote Dickinson (1973): "Thus we resorted to an entirely numerical approach for obtaining a Newtonian cooling coefficient $a_0(z)$ for small departures from the reference temperature profile $T_0(z)$. That is, if $Q(T)$ is the infrared cooling rate for a temperature profile T(z), then
$$a_0(z) = \frac{Q(T_0+\delta)-Q(T_0-\delta)}{2\delta}$$
where $\delta$ is a small temperature perturbation (we used $\delta = 0.1^o K$)."

Assuming $Q(T)$ for $1 \times CO_2$ or $2 \times CO_2$ is a smooth function, we can infer that if $Q(T)$ for $2 \times CO_2$ at some altitude level is approximately 50% larger than that for $1 \times CO_2$ at the same altitude level, then $a_0(z)$ for the former is also approximately 50% larger than that for the latter at that altitude level. Note that the shape of the profile Q(z) in Fig. 1 depicted by Dickinson (1973) is very similar to that of the profile $a_0(z)$ in his Fig. 3. In other words, the value of $a_0(z)$ is approximately proportional to that of Q(z).

In addition, we don't need to know how radiative equilibrium temperatures change in response to increasing $CO_2$ concentration when we study how the wave-mean flow interactions generate the QBO, because the temperature fields associated with atmospheric waves relax back to the zonal mean temperatures rather than radiative equilibrium temperatures.

Finally, Dickinson (1973) implied that below the 0.2 hPa level the value of an estimated Newtonian cooling coefficient $a(z)$ is not sensitive to how a temperature profile $T(z)$ is chosen (refer to his Eqs. (2) and (3)).

The usage of the Plass value of 50% should be either (1) justified in light of these considerations or (2) an alternative reliable estimate should be provided of the radiative damping rate response to CO2 doubling (an order of magnitude estimate is fine). If no projection of radiative damping rate with CO2 doubling exists in the literature, then one should be produced (e.g. using a radiative transfer model). Such a projection of radiative damping rate, necessary for the arguments in the paper, would constitute a valuable contribution in its own right.

The maximum value we used is **30%** rather than the Plass value of 50% (refer to lines 215-219 in the revised manuscript). Plass (1956) claimed "The probable error of the cooling rate is estimated by introducing arbitrary variations into the original transmission functions and calculating their influence on the final result. The probable error obtained in this manner is about 10 per cent below 20 km, increasing to 30 per cent at 50 km and becoming rather uncertain above 60 km. Again, the relative differences between the various curves should be considerably more accurate than their magnitude."

Even if we regard the probable error of the cooling rate is 30 percent, the relative differences between the various cooling rates calculated by Plass (1956) should be considerably smaller 30 percent. Since the relative differences in cooling rates calculated by Plass (1956) are around 50% above 35 km, our value of 30% is smaller than the lower bound of uncertainty, i.e., 50% - 50%*30% = 35%. In other words, our choice of 30% is a conservative estimate.

Second, there is tension in the manuscript between the interpretability and predictive value of the results. Using the 1D model makes a strong decision in favor of interpretability, which is well justified

by the approach and the results. Noting that the QBO period in the basic state and the response to changes in radiative damping are both highly sensitive to minor changes in the model formulation, it appears that the 1D model can provide, at best, the sign and order of magnitude of the period change in response to an increase in radiative damping. Therefore, modeling decisions that sacrifice the interpretability of the final results without impacting the sign or order of magnitude of the final result should be justified or avoided. Two example decisions in the paper are as follows. For each, the sacrifice of interpretability should be (1) justified in terms of predictive value or some other objective or (2) the simpler case should be considered:

• The Holton (1972) formulation used in this paper is driven by asymmetric waves (a Kelvin wave and a Rossby wave with different dispersion relations and wavenumbers). The Plumb (1977) formulation is driven by symmetric wave stress (equal and opposite gravity waves), allowing for clear interpretation of the model dynamics in terms of a small number of dimensionless parameters. Is there a benefit to using asymmetric wave forcing in this paper that justifies it at the cost of sacrificing the interpretability of the symmetric formulation?

Plumb (1977) provided a simpler and elegant theoretical framework to illuminate the essence of the QBO. What is more, Plumb (1977) paved the way for the experimental tour de force (Plumb and McEwan 1978) guided by "a small number of dimensionless parameters". Plumb and McEwan (1978) demonstrated how a standing internal wave with sufficiently large amplitudes forced at the lower boundary of an annulus of salt-stratified water generated an oscillatory mean flow with relatively long periods compared to the period of the internal wave. This incarnation of the QBO analog cleared up any lingering doubts about the theory of wave-mean flow interactions.

However, the Plumb (1977) formulation is best suited to study non-rotating systems such as those in laboratories rather than the planetary-scale rotating Earth system. Although some authors used it to illustrate the stratospheric QBO by introducing a Kelvin wave and an "anti-Kelvin wave", there is no "anti-Kelvin wave" in the terrestrial atmosphere. The Holton (1972) formulation was based on the observations that planetary-scale waves in the Equatorial lower stratosphere are dominated by Kelvin waves of zonal wavenumber 1-2 and mixed Rossby-gravity waves of wavenumber 4 (Andrews et al. 1987).

• Another source of the tension is in the choice to include changes in the buoyancy frequency N in the projections of the model response to radiative damping changes. Including the small (2.5%) changes in N seems to be so marginal that its effects are primarily to sacrifice interpretability without a clear benefit. It can still be useful to include a sensitivity study to changes in N, but this sensitivity study should be distinguished from the main line of argumentation in the paper. Note that in the 1D model, the buoyancy frequency N and radiative damping µ are always multiplied together, such that their combined effects are tantamount to considering a $(1.5 * 1.025 \rightarrow) \approx 54\%$ change in radiative damping (or buoyancy frequency) alone, not significantly different than the 50% change in radiative damping.

Following your suggestion by excluding the small changes in N, we have redone a lot of experiments. Subsequently, figures 3, 5, and 6 have been re-plotted and the manuscript has been revised accordingly.

**3  Minor comments**

It would be appreciated if the following minor comments regarding the content and structure of the paper were addressed:

• The decomposition of $\bar{u} = \bar{u}_{sa} + \bar{u}_{QBO}$ in equation (5) gives the impression that the evolution of $\bar{u}_{QBO}$ depends only on $\bar{u}_{QBO}$, and that the influence of $\bar{u}_{sa}$ has been factored out. Yet $\bar{u}_{sa}$ still impacts the evolution of $\bar{u}_{QBO}$ through the wave forcing $\bar{F}_i$. Because $\bar{F}_i$ is nonlinear in the zonal wind profile, the SAO will impact the wave forcing, which changes the mean wind and then alters the diffusion profile. So, the dynamics are not simply resulting from the sum of a linear QBO dynamic and a linear SAO dynamic. Given the limitations of this decomposition, what benefit is provided by its inclusion?

Yes, we agree with your reasoning and expected that the simulated QBO periods should be sensitive to $G$, the imposed semiannual forcing. However, the simulated QBO periods are not sensitive to the imposed semiannual forcing provided that $G$ does not exceed the values employed by HL (refer to lines 314-316 of the version with track changes).
We used "the decomposition of $\bar{u} = \bar{u}_{sa} + \bar{u}_{QBO}$ in equation (5)" to highlight this bizarre behavior and further illustrated that the deficiency is closely related to the insufficient height of the model top.

• Lines 301-312 list all numerical values from Figure 6. Is it necessary to list all numerical values when the figure provides their approximate value? If so, then perhaps a table can be supplied instead of the figure or in the supplementary information. I suspect that the relevant information from Figure 6 can be conveyed in a more concise way.

We have eliminated this unpleasant verbosity.

• In the Introduction, the following question is raised: "Does the competing effect between [upwelling and enhanced wave stress] leave some room for increasing stratospheric radiative damping to exert an influence on the QBO period?" (Line 137) In the Conclusions, comprehensive QBO models are noted to have significant variance in their projections of period. A quantitative comparison would be useful here; allowing that the 1D model provides at best an order of magnitude estimate, is there reason to believe that period changes in GCMs are small enough that radiative damping could potentially impact their sign? Their magnitude? Or on the contrary, are the period changes in GCMs large enough in magnitude that radiative damping would not be expected to have a significant bearing on the sign or magnitude of the change?

We are not in position to answer those important questions. This report aims to provoke the readers to think and/or conduct more researches to answer them.

• Lines 343 - 345: Recently, doubt has been cast on the role of upwelling on QBO amplitude (Match and Fueglistaler, 2019, 2020). A more nuanced assessment would be appreciated than "The amplitude decrease is associated with a strengthened residual mean circulation, also consistent with the literature, although the vertical structure of the circulation response is nontrivial."

Done.

**4 Edits**

(166) "mixing Rossby-gravity wave" should be "mixed Rossby-gravity wave"

Done.

**References**

Andrews, D. G., Holton, J. R., and Leovy, C. B.: Middle Atmosphere Dynamics, Academic Press, 489 pp, 1987.

Dickinson, R. E.: Method of parameterization for infrared cooling between altitudes of 30 and 70 kilometers, J. Geophys. Res., 78, 4451–4457, https://doi.org/10.1029/JC078i021p04451, 1973.

Plass, G. N.: The influence of the 15μ carbon-dioxide band on the atmospheric infra-red cooling rate, Quart. J. Roy. Meteor. Soc., 82, 310–324, https://doi.org/10.1002/qj.49708235307, 1956.

Plumb, R. A.: The interaction of two internal waves with the mean flow: Implications for the theory of the quasi-biennial oscillation, J. Atmos. Sci., 34, 1847–1858, https://doi.org/10.1175/1520-0469(1977)034<1847:TIOTIW>2.0.CO;2, 1977.

Plumb, R. A., and McEwan, A. D.: The instability of a forced standing wave in a viscous stratified fluid: A laboratory analogue of the quasi-biennial oscillation. J. Atmos. Sci., 35, 1827–1839, https://doi.org/10.1175/1520-0469(1978)035<1827:TIOAFS>2.0.CO;2, 1978.

---

## Author Comment (AC5) · 5 Dec 2020

**Response to Reviewer 2**

This is the first review of the manuscript "The Impact of Increasing Stratospheric Radiative Damping on the QBO Period" by Tiehan Zhou, Kevin DallaSanta, Larissa Nazarenko, Gavin A. Schmidt, Paper # acp-2020-925. The approach of the paper is to conduct sensitivity experiments using 1-D mechanistic model to find an impact of radiative dumping in the stratosphere to QBO period. Experimental parameters in this paper are Newtonian cooling (alpha; unit:s-1)/Brunt-Vaisala frequency (N; unit: s-1), and the upper boundary conditions (G). Diagnostics are monthly zonal wind, and the frequency power spectra using the fast Fourier transform. The results are interesting and relevant to Atmospheric Chemistry and Physics because this paper is trying to ascertain the trend of the QBO period in a warming climate, focusing on the radiative damping that would influence that period. But I suggest a major substantial revision since I have serious concerns about generalization and validation of authors' results. (mainly described in the line-by-line comments).

We thank you for your helpful comments and suggestions and will address them point by point as below.

Major comments
1. Statistical significance L047 "the doubling of CO2 shortens the QBO period by 4.7%" What kind of meaning does a value of 4.7 hold for the science? Results of sensitivity experiments using 1D model mostly depend on an assumption of the experimental design, here in Newtonian cooling profiles of Fig. 1. Can you show realistic vertical profiles of Newtonian cooling with standard deviations? And then, you can estimate errors about the shortening of the QBO period, from an assumption of errors of Newtonian cooling.

The reviewer 1 suggested that we should focus on the interpretation of the hypothesized mechanism and its attendant uncertainties without asking too much of its predictive value. Thus, the numbers such as 4.7% are used for qualitative interpretation rather than quantitative prediction. For example, in Section 4, We found that when the Brunt-Väisälä frequency was decreased by 2.5%, the simulated QBO period was slightly lengthened from 30 months to 30.2 months; we also found that when the scale height in the stratosphere was decreased by 2.3%, the simulated QBO period was shortened from 30 months to 29.3 months. Those findings only suggest that the increase of the stratospheric radiative damping contributes more to the shortening of the QBO period than the shrinkage of scale height in the stratosphere while the contribution of the change in the Brunt-Väisälä frequency is almost negligible. By the way, we also added into the manuscript some discussions on the uncertainties in the relative change of cooling coefficient.

2. Scale height Scale height would be changed in a warming climate. How does the change of scale height due to the temperature change in a warming climate affect the QBO period?

An extra sensitivity test has been conducted. Accordingly, a paragraph has been added in the manuscript (refer to lines 409-417 of the revised version with track changes).

3. Ozone The ozone also affects QBO period. It is useful for readers to assess an effect of the ozone on the QBO period using 1D model in a warming climate. Shibata, K., and M. Deushi (2005), Radiative effect of ozone on the quasi-biennial oscillation in the equatorial stratosphere, Geophys. Res. Lett., 32, L24802, doi:10.1029/2005GL023433.

Yes, ozone does affect the QBO period, and we are studying this issue. This paper addresses the impact of the doubling of $CO_2$ concentration on the QBO period from the viewpoint of increasing stratospheric radiative damping. According to the latest Scientific Assessment of Ozone Depletion

completed in 2018, ozone will heal completely before the $CO_2$ concentration is doubled. Thus, we chose not to deal with ozone in this study.

L10-16. Why the authors do not show that QBOs simulated in this paper are derived only from planetary waves and that gravity waves are not included? Without manifestation about exploring the response to enhancing radiative damping of only planetary waves in the experiments, their conclusion would lead to misleading.

Richter et al. (2020) pointed out that the largest uncertainty in the response of the QBO in a warming climate comes from the representation of parameterized gravity waves in climate models. Section 5 suggested that high-resolution models such as those used by Kawatani et al. (2011, 2019) be used to further our understanding.

L15-16. Most climate models project a strengthening of tropical vertical residual velocity, as you mentioned in the introduction. This could contribute to projecting a lengthening of the QBO period, which is opposite direction to the authors' conclusion.

As discussed in Sections 1 and 5 of the manuscript, our focus is placed on how the physical processes other than wave momentum flux entering the stratosphere and tropical vertical residual velocity could exert an influence on the trend of the QBO period in a warming climate. This report doesn't intend to answer the question: Will the QBO period ultimately become longer or shorter in the warming climate?

L232. "QBO was not essential for QBO theory" You can estimate QBO power with/without SAO. To what extent does the SAO impact the QBO power spectrum?

QBO and SAO power spectra were not estimated separately. They are plotted separately for the sake of visual effects. As per Plumb (1977), the SAO exerts little, if any, influence on the QBO power spectrum in this kind of model configurations.

Minor comments L301-323. Redundant descriptions. You can omit most of them.

The redundancy has been eliminated.

**References**

Kawatani, Y., Hamilton, K., and Watanabe, S.: The quasi-biennial oscillation in a double CO2 climate, J. Atmos. Sci., 68, 265–283, https://doi.org/10.1175/2010JAS3623.1, 2011.

Kawatani, Y., Hamilton, K., Sato, K., Dunkerton, T. J., Watanabe, S., and Kikuchi, K.: ENSO Modulation of the QBO: Results from MIROC Models with and without Nonorographic Gravity Wave Parameterization, J. Atmos. Sci., 76, 3893–3917, https://doi.org/10.1175/JAS-D-19-0163.1, 2019.

Plumb, R. A.: The interaction of two internal waves with the mean flow: Implications for the theory of the quasi-biennial oscillation, J. Atmos. Sci., 34, 1847–1858, https://doi.org/10.1175/1520-0469(1977)034<1847:TIOTIW>2.0.CO;2, 1977.

Richter, J. H., Butchart, N., Kawatani, Y., Bushell, A. C., Holt, L., Serva, F., Anstey, J., Simpson, I. R., Osprey, S., Hamilton, K., Braesicke, P., Cagnazzo, C., Chen, C.-C., Garcia, R. R., Gray, L. J., Kerzenmacher, T., Lott, F., McLandress, C., Naoe, H., Scinocca, J., Stockdale, T. N, Versick, S., Watanabe, S., Yoshida, K., Yukimoto, S.: Response of the Quasi-Biennial Oscillation to a warming climate in global climate models, Q. J. R. Meteorol. Soc., 1–29. https://doi.org/10.1002/qj.3749, 2020.

---

## Referee Report (RR1)

**Second Review of Manuscript submitted to ACP: "The Impact of Increasing Stratospheric Radiative Damping on the QBO Period" by Zhou et al (2020)**

December 26, 2020

**Suggestion: Major revisions**

**1 Overview**

Here, I review the second submitted version of "The Impact of Increasing Stratospheric Radiative Damping on the QBO Period" by Zhou et al (2020). As in the first submitted manuscript, this paper proposes a new radiative-dynamical mechanism that could have implications for the future properties of the QBO. This paper proposes that increasing $CO_2$ is expected to increase the radiative damping rate in the stratosphere. Using an idealized 1D model of the QBO, this increase in the radiative damping rate would be expected to modestly shorten the period of the QBO.

I will refer to the first reviewed manuscript as V1 and the most recently submitted manuscript as V2. V2 is similar in scientific content and presentation to V1, with the exception of specific revisions made in light of reviewer comments. In my previous review of V1, I identified two items that required major revision: (1) the tension between interpretability and predictability and (2) the physical justification for using the Plass (1956) result to estimate future radiative damping. V2 has satisfactorily resolved the tension between interpretability and predictability, as will be elaborated upon further. Neither V2 nor the specific response to my previous review has satisfactorily resolved my concerns about the physical justification for using the Plass (1956) result, as will be elaborated upon further, and which leads me to recommend Major Revisions.

**2 Major Revisions from V1**

**2.1 Resolved in V2:**

In reviewing V1, a tension between predictability and interpretability of the main results was identified. This tension resulted from the addition of realistic but minor predictive elements to the key simulations in the paper, which made it more difficult to isolate the QBO period changes owing to the proposed mechanism versus to the other minor elements (e.g. changes in buoyancy frequency). This tension has been largely resolved in V2, which focuses on the interpretability of the results and considers minor predictive elements only in the Discussion. It is clear in V2 that the minor predictive elements in V1 were not leading to significant changes in the basic QBO behavior (the $N^2$ changes alone appear to only change the QBO period from 30 months to 30.2 months on line 348). I am satisfied with this resolution of the tension between predictability and interpretability in favor of emphasizing interpretability.

**2.2 Requires major revision in V2:**

In reviewing V2, I commented that the usage of a 50% change in radiative damping rate in response to increasing CO2 required further justification. It was unclear whether the Plass (1956) result was providing the appropriate justification. After reviewing the response to my review of V1 and reading V2, I remain concerned about the justification for the radiative damping changes used in the paper. I will elaborate on my concerns below so as to be clear about the source of my confusion.

If the radiative heating rate represents a linear damping with rate $\alpha$ [s$^{-1}$] of temperature $T$ [K] relative to the local radiative equilibrium temperature $T_E$ [K], then the radiative heating rate $Q$ [K s$^{-1}$] obeys the Newtonian cooling equation:

$$Q = \alpha(T_E - T) \tag{1}$$

The accepted method for diagnosing radiative damping rate is to consider two states S1 and S2 where all atmospheric properties are held fixed except that temperatures are prescribed to vary between S1 and S2 and the radiative heating rate responds to the temperature profile. The radiative heating rates are calculated using a radiative transfer model. If the radiative heating rates satisfy the assumption of linearity, therefore obeying equation 1, then states S1 and S2 separately obey the following equations:

$$Q_1 = \alpha(T_E - T_1)$$

$$Q_2 = \alpha(T_E - T_2)$$

The above equation system has two equations in two unknowns (unknowns are $\alpha$ and $T_E$), so it is possible to solve for the unknowns. Solving for $\alpha$ yields:

$$\alpha = -\frac{Q_2 - Q_1}{T_2 - T_1} \tag{2}$$

Note that the above Equation 2 is equivalent to Dickinson's Equation 1. Note that it is only possible to solve for $\alpha$ if $\alpha$ and $T_E$ are assumed to be constant between states S1 and S2. The assumption of constant $\alpha$ and $T_E$ in S1 and S2 is reasonable because $\alpha$ and $T_E$ are thought of as functions of the atmospheric composition, which was held fixed.

Unlike in this typical approach, the experiments in Plass (1956) consider the radiative heating rate [K s$^{-1}$] in an atmosphere with temperature held fixed but composition varied. When atmospheric composition is varied, it is not realistic to assume that $\alpha$ and $T_E$ remain constant. Two states from the Plass (1956) approach, denoted S1$'$ and S2$'$, obey the following equations:

$$Q_1' = \alpha_1'(T_{E,1}' - T')$$

$$Q_2' = \alpha_2'(T_{E,2}' - T')$$

The above two equations have four unknowns ($\alpha_i'$ and $T_{E,i}'$ for i = 1,2). Therefore, $\alpha_i'$ and $T_{E,i}'$ are underdetermined. Adding experiments calculated with new concentrations of $CO_2$ would not make the equation set determined, because each additional equation adds two unknowns $\alpha_i'$ and $T_{E,i}'$. Therefore, it is not possible to solve for $\alpha_i'$ or $T_{E,i}'$, nor is it possible to solve for $\alpha_1'/\alpha_2'$.

If it is assumed that there is constant $T_E'$ in the Plass (1956) experiments, the equation set now has three unknowns, and it is possible to solve for $\alpha_1'/\alpha_2'$ as follows:

$$\frac{\alpha_1'}{\alpha_2'} = \frac{Q_1'}{Q_2'}$$

V2 appears to use the above equality between the ratio of heating rates and ratio of radiative damping rates to justify the use of Plass (1956) to project the magnitude of future change in radiative damping rate. However, the above equality depends on the assumption that $T_E$ is constant as $CO_2$ varies, an assumption which does not appear justifiable given that increasing $CO_2$ is expected to decrease $T_E$ in the stratosphere (Manabe et al., 1967). Without assuming constant $T_E$, it appears that the changes in radiative heating rate cannot be used to constrain the changes in $\alpha$.

Knowing that $T_E$ changes in response to increasing $CO_2$, one could alternatively assume that $\alpha$ stays constant with increasing $CO_2$ while $T_E$ changes, which would result in the following relationship:

$$\frac{T_{E,1}' - T'}{T_{E,2}' - T'} = \frac{Q_1'}{Q_2'}$$

In reality, there are probably changes in both $T_E$ and $\alpha$ as $CO_2$ concentrations are changed. These changes must be distinguished, and the component stemming from a change in $\alpha$ isolated, in order to obtain an order of magnitude estimate for the period change resulting from this mechanism. The 50% change in radiative heating rates from Plass (1956) in response to changes in $CO_2$ does not necessitate that the changes in radiative damping rate are $O(50\%)$, as they could be much smaller (or larger).

In light of these concerns, the substance of my recommendation for major revisions from my previous review still stands: The usage of the Plass value of 50% [or 30% in V2] should be either (1) justified in light of these considerations or (2) an alternative reliable estimate should be provided of the radiative damping rate response to CO2 doubling (an order of magnitude estimate is fine). If no projection of radiative damping rate with CO2 doubling exists in the literature, then one should be produced (e.g. using a radiative transfer model). Such a projection of radiative damping rate, necessary for the arguments in the paper, would constitute a valuable contribution in its own right.

**3   Minor comments**

- It is stated that the semi-annual oscillation profile has zero curvature, so there is no diffusion of that profile (Line 263). However, there is a kink in $u_{SA}$ at 28 km at the intersection of the constant (zero) value below and the linear profile above. The curvature at the kink is large and undefined, so the statement $\partial^2 \bar{u}_{SA}/\partial z^2 = 0$ on Line 263 is not accurate. I would intuitively expect diffusion across this kink. The omission of the diffusion acting on $\bar{u}_{SA}$ from equation (5) therefore needs to be justified.

- In light of the considerations in the previous item, but also more broadly, I remain unsure about what is gained by the decomposition of $\bar{u} = \bar{u}_{QBO} + \bar{u}_{SA}$. Given that the wave driving acts on both $u_{QBO}$ and $u_{SA}$ and (as argued above) the diffusion acts on both $u_{QBO}$ and $u_{SA}$, it is unclear whether $u_{QBO}$ can be fruitfully isolated and treated in a separate analytical framework.

- Line 296 states that "unphysical behavior" arises when the period does not change smoothly as a function of the magnitude of $G$. It is not clear to me why this behavior is unphysical. On the one hand, it is conceivable for $G$ to be weak enough that it would not impact the period of the QBO, and therefore the period of the QBO would match the $G = 0$ state. It is also conceivable that there is phase-locking between the semi-annual oscillation and the QBO, such that if $G$ is varied over a small enough set of values, then the SAO might maintain the same phase relationship with the QBO, and therefore not exhibit a period response. (In a totally different context, such phase-locking across a range of parameter values of the QBO was reported in response to upwelling variations in Rajendran et al. (2016).) In light of these plausible

explanations, I recommend that either some insight be gained regarding the phase locking, or the language "unphysical behavior" be softened, e.g. to "unexpected behavior".

- Line 367 provides an estimate for the shortening of the QBO period stemming from the doubling of CO2 of $7.4\% \pm 2.5\%$. The error estimate of $2.5\%$ was estimated by propagating through the $30\%$ error estimate from Plass (1956) on the radiative damping rate. This error statement is therefore a statement on the estimated error in QBO period change *owing to errors in the radiative damping rate.* As presented, the error estimate appears to be a more comprehensive statement concerning error in the 1D results. However, V2 presented substantial systemic uncertainty, another source of error. It was shown that the period change was $15\%$ when using the Plumb model, which is a plausible formulation of the problem and far outside the error margin on Line 367. I recommend that the narrow scope of the error margin on Line 367 be stated, by qualifying that they refer only to expected errors from uncertainty in radiative damping rate. The much larger error owing to the formulation of the QBO model should be noted in any statement regarding uncertainty in the magnitude of the period change. (Note: The error estimate will ultimately need to be consistent with the resolution of my major comment regarding the estimate of the radiative damping rate.)

**References**

Manabe, S., R. T. Wetherald, S. Manabe, and R. T. Wetherald, 1967: Thermal Equilibrium of the Atmosphere with a Given Distribution of Relative Humidity. *Journal of the Atmospheric Sciences*, **24 (3)**, 241–259, doi:10.1175/1520-0469(1967)024⟨0241:TEOTAW⟩2.0. CO;2, URL http://journals.ametsoc.org/doi/abs/10.1175/1520-0469{\%}25281967{\%} }2529024{\%}253C0241{\%}253ATEOTAW{\%}253E2.0.CO{\%}253B2.

Plass, G. N., 1956: The influence of the 15p carbon-dioxide band on the atmospheric infra-red cooling rate. Tech. rep. doi:10.1002/qj.49708235307.

Rajendran, K., I. M. Moroz, P. L. Read, and S. M. Osprey, 2016: Synchronisation of the equatorial QBO by the annual cycle in tropical upwelling in a warming climate. *Quarterly Journal of the Royal Meteorological Society*, **142 (695)**, 1111–1120, doi:10.1002/qj.2714, URL http://doi.wiley.com/10.1002/qj.2714.

---

## Author Response (AR2)

**Response to Reviewer 1**

**Suggestion: Major revisions**

**1 Overview**

Here, I review the second submitted version of "The Impact of Increasing Stratospheric Radiative Damping on the QBO Period" by Zhou et al (2020). As in the first submitted manuscript, this paper proposes a new radiative-dynamical mechanism that could have implications for the future properties of the QBO. This paper proposes that increasing $CO_2$ is expected to increase the radiative damping rate in the stratosphere. Using an idealized 1D model of the QBO, this increase in the radiative damping rate would be expected to modestly shorten the period of the QBO.

I will refer to the first reviewed manuscript as V1 and the most recently submitted manuscript as V2. V2 is similar in scientific content and presentation to V1, with the exception of specific revisions made in light of reviewer comments. In my previous review of V1, I identified two items that required major revision: (1) the tension between interpretability and predictability and (2) the physical justification for using the Plass (1956) result to estimate future radiative damping. V2 has satisfactorily resolved the tension between interpretability and predictability, as will be elaborated upon further. Neither V2 nor the specific response to my previous review has satisfactorily resolved my concerns about the physical justification for using the Plass (1956) result, as will be elaborated upon further, and which leads me to recommend Major Revisions.

We are grateful for your thoughtful and thorough review, and regretful for the inadequacy of our previous revision. Hopefully, this version of manuscript has adequately addressed your concerns. Note that all line numbers refer to those in the tracked version.

**2 Major Revisions from V1**

**2.1 Resolved in V2:**

In reviewing V1, a tension between predictability and interpretability of the main results was identified. This tension resulted from the addition of realistic but minor predictive elements to the key simulations in the paper, which made it more difficult to isolate the QBO period changes owing to the proposed mechanism versus to the other minor elements (e.g., changes in buoyancy frequency). This tension has been largely resolved in V2, which focuses on the interpretability of the results and considers minor predictive elements only in the Discussion. It is clear in V2 that the minor predictive elements in V1 were not leading to significant changes in the basic QBO behavior (the $N^2$ changes alone appear to only change the QBO period from 30 months to 30.2 months on line 348). I am satisfied with this resolution of the tension between predictability and interpretability in favor of emphasizing interpretability.

We are glad that this segment of our revision is satisfactory.

**2.2 Requires major revision in V2:**

In reviewing V2, I commented that the usage of a 50% change in radiative damping rate in response to increasing $CO_2$ required further justification. It was unclear whether the Plass (1956) result was providing the appropriate justification. After reviewing the response to my review of V1 and reading V2, I remain

concerned about the justification for the radiative damping changes used in the paper. I will elaborate on my concerns below so as to be clear about the source of my confusion.

If the radiative heating rate represents a linear damping with rate $\alpha$ [s$^{-1}$] of temperature $T$ [K] relative to the local radiative equilibrium temperature $T_E$ [K], then the radiative heating rate $Q$ [K s$^{-1}$] obeys the Newtonian cooling equation:

$$Q = \alpha(T_E - T) \tag{1}$$

The accepted method for diagnosing radiative damping rate is to consider two states S1 and S2 where all atmospheric properties are held fixed except that temperatures are prescribed to vary between S1 and S2 and the radiative heating rate responds to the temperature profile. The radiative heating rates are calculated using a radiative transfer model. If the radiative heating rates satisfy the assumption of linearity, therefore obeying equation 1, then states S1 and S2 separately obey the following equations:

$$Q_1 = \alpha(T_E - T_1)$$

$$Q_2 = \alpha(T_E - T_2)$$

The above equation system has two equations in two unknowns (unknowns are $\alpha$ and $T_E$), so it is possible to solve for the unknowns. Solving for $\alpha$ yields:

$$\alpha = \frac{Q_2 - Q_1}{T_2 - T_1} \tag{2}$$

Note that the above Equation 2 is equivalent to Dickinson's Equation 1. Note that it is only possible to solve for $\alpha$ if $\alpha$ and $T_E$ are assumed to be constant between states S1 and S2. The assumption of constant _ and TE in S1 and S2 is reasonable because $\alpha$ and $T_E$ are thought of as functions of the atmospheric composition, which was held fixed.

Unlike in this typical approach, the experiments in Plass (1956) consider the radiative heating rate [K s$^{-1}$] in an atmosphere with temperature held fixed but composition varied. When atmospheric composition is varied, it is not realistic to assume that $\alpha$ and $T_E$ remain constant. Two states from the Plass (1956) approach, denoted $S1'$ and $S2'$, obey the following equations:

$$Q_1' = \alpha_1'(T_{E,1}' - T')$$

$$Q_2' = \alpha_2'(T_{E,2}' - T')$$

The above two equations have four unknowns ($\alpha_i'$ and $T_{E,i}'$ for I =1, 2). Therefore, $\alpha_i'$ and $T_{E,i}'$ are underdetermined. Adding experiments calculated with new concentrations of $CO_2$ would not make the equation set determined, because each additional equation adds two unknowns $\alpha_i'$ and $T_{E,i}'$. Therefore, it is not possible to solve for $\alpha_i'$ and $T_{E,i}'$, nor is it possible to solve for $\alpha_1'/\alpha_2'$.

If it is assumed that there is constant $T_E'$ in the Plass (1956) experiments, the equation set now has three unknowns, and it is possible to solve for $\alpha_1'/\alpha_2'$ as follows:

$$\frac{\alpha_1'}{\alpha_2'} = \frac{Q_1'}{Q_2'}$$

V2 appears to use the above equality between the ratio of heating rates and ratio of radiative damping rates to justify the use of Plass (1956) to project the magnitude of future change in radiative damping rate. However, the above equality depends on the assumption that $T_E$ is constant as $CO_2$ varies, an assumption which does not appear justifiable given that increasing $CO_2$ is expected to decrease $T_E$ in the stratosphere (Manabe et al., 1967). Without assuming constant $T_E$, it appears that the changes in radiative heating rate cannot be used to constrain the changes in $\alpha$.

Knowing that $T_E$ changes in response to increasing $CO_2$, one could alternatively assume that $\alpha$ stays constant with increasing $CO_2$ while $T_E$ changes, which would result in the following relationship:

$$\frac{T'_{E,1} - T'}{T'_{E,2} - T'} = \frac{Q'_1}{Q'_2}$$

In reality, there are probably changes in both $T_E$ and $\alpha$ as $CO_2$ concentrations are changed. These changes must be distinguished, and the component stemming from a change in $\alpha$ isolated, in order to obtain an order of magnitude estimate for the period change resulting from this mechanism. The 50% change in radiative heating rates from Plass (1956) in response to changes in $CO_2$ does not necessitate that the changes in radiative damping rate are O(50%), as they could be much smaller (or larger).

In light of these concerns, the substance of my recommendation for major revisions from my previous review still stands: The usage of the Plass value of 50% [or 30% in V2] should be either (1) justified in light of these considerations or (2) an alternative reliable estimate should be provided of the radiative damping rate response to $CO_2$ doubling (an order of magnitude estimate is fine). If no projection of radiative damping rate with $CO_2$ doubling exists in the literature, then one should be produced (e.g., using a radiative transfer model). Such a projection of radiative damping rate, necessary for the arguments in the paper, would constitute a valuable contribution in its own right.

We thank you for your thoughtful and thorough explanations. Accordingly, we used a sophisticated radiative transfer model to evaluate how the Newtonian cooling coefficients change in response to the doubling of $CO_2$ concentration, which is described in lines 208-239 of the latest version of the manuscript.

**3 Minor comments**

- It is stated that the semi-annual oscillation profile has zero curvature, so there is no diffusion of that profile (Line 263). However, there is a kink in $u_{SA}$ at 28 km at the intersection of the constant (zero) value below and the linear profile above. The curvature at the kink is large and undefined, so the statement $\frac{\partial^2 \bar{u}_{SA}}{\partial z^2} = 0$ on Line 263 is not accurate. I would intuitively expect diffusion across this kink. The omission of the diffusion acting on $\bar{u}_{SA}$ from equation (5) therefore needs to be justified.

We thank you for your rigorous insight. Subsequently, we have removed Eq. (5) and revised that section.

- In light of the considerations in the previous item, but also more broadly, I remain unsure about what is gained by the decomposition of $\bar{u} = \bar{u}_{QBO} + \bar{u}_{SA}$. Given that the wave driving acts on both $\bar{u}_{QBO}$ and $\bar{u}_{SA}$ and (as argued above) the diffusion acts on both $\bar{u}_{QBO}$ and $\bar{u}_{SA}$, it is unclear whether $\bar{u}_{QBO}$ can be fruitfully isolated and treated in a separate analytical framework.

As mentioned above, we have removed that unfruitful segment.

- Line 296 states that "unphysical behavior" arises when the period does not change smoothly as a function of the magnitude of G. It is not clear to me why this behavior is unphysical. On the one hand, it is conceivable for G to be weak enough that it would not impact the period of the QBO, and therefore the period of the QBO would match the G = 0 state. It is also conceivable that there is phase-locking between the semi-annual oscillation and the QBO, such that if G is varied over a

small enough set of values, then the SAO might maintain the same phase relationship with the QBO, and therefore not exhibit a period response. (In a totally different context, such phase-locking across a range of parameter values of the QBO was reported in response to upwelling variations in Rajendran et al. (2016).) In light of these plausible explanations, I recommend that either some insight be gained regarding the phase locking, or the language "unphysical behavior" be softened, e.g., to "unexpected behavior".

We have changed "unphysical behavior" into "unexpected behavior" as suggested.

- Line 367 provides an estimate for the shortening of the QBO period stemming from the doubling of $CO_2$ of 7.4% $\pm$ 2.5%. The error estimate of 2.5% was estimated by propagating through the 30% error estimate from Plass (1956) on the radiative damping rate. This error statement is therefore a statement on the estimated error in QBO period change owing to errors in the radiative damping rate. As presented, the error estimate appears to be a more comprehensive statement concerning error in the 1D results. However, V2 presented substantial systemic uncertainty, another source of error. It was shown that the period change was 15% when using the Plumb model, which is a plausible formulation of the problem and far outside the error margin on Line 367. I recommend that the narrow scope of the error margin on Line 367 be stated, by qualifying that they refer only to expected errors from uncertainty in radiative damping rate. The much larger error owing to the formulation of the QBO model should be noted in any statement regarding uncertainty in the magnitude of the period change. (Note: The error estimate will ultimately need to be consistent with the resolution of my major comment regarding the estimate of the radiative damping rate.)

We have added "and taking into account the uncertainty due to the radiative damping rate" in line 442.

**References**

Manabe, S. and Wetherald, R. T.: Thermal equilibrium of the atmosphere with a given distribution of

relative humidity, J. Atmos. Sci., 24, 241–259, https://doi.org/10.1175/1520-0469(1967)024%3C0241:TEOTAW%3E2.0.CO;2, 1967.

Plass, G. N.: The influence of the 15µ carbon-dioxide band on the atmospheric infra-red cooling rate, Quart. J. Roy. Meteor. Soc., 82, 310–324, https://doi.org/10.1002/qj.49708235307, 1956.

Rajendran, K., Moroz, I. M., Read, P. L., and Osprey, S. M.: Synchronisation of the equatorial QBO by the annual cycle in tropical upwelling in a warming climate, Q. J. Roy. Meteor. Soc., 142, 1111–1120, https://doi.org/10.1002/qj.2714, 2016.

**Response to Reviewer 2**

**This is the review of the revised manuscript "The Impact of Increasing Stratospheric Radiative Damping on the QBO Period"**

The results are interesting and relevant to Atmospheric Chemistry and Physics. However, I do not understand their explanations about ozone effect on the QBO period, so I recommend a minor revision.

We appreciate you for your helpful comments and will address this specific issue below. Note that all line numbers refer to those in the tracked version.
* * *
>> 2. Scale height.
I understand that the sensitivity of scale height is smaller than that of the radiative dumping.

We thank you for your approval.

>> 3. Ozone
>ozone does affect the QBO period,
> According to the latest Scientific Assessment of Ozone Depletion completed in 2018, ozone will heal completely before the $CO_2$ concentration is doubled.

We completely agree with you.

I think without ozone depletion by ODS, ozone at above 20 hPa does affect the QBO because in this region ozone is dominated by temperature-dependent photochemistry.

We have added one paragraph (lines 405-415) and slightly modified line 435 to address this issue.

SD2005 indicated that the QBO with non-interactive (interactive) ozone reproduces a 27-month (31-month) period, which results in increase by 15%. This amount is much larger amount than the QBO shortening by 7.4% due to enhanced radiative damping of your finding.

It is an interesting and puzzling research topic which is beyond the scope of this paper due to the following two reasons.

(1) SD2005, i.e., Shibata and Deushi (2005) reported that "Two control runs are designed to simulate the QBO with realistic period of the oscillation: the one with non-interactive ozone reproducing a 27-month period and the other with interactive ozone yielding a 31-month period. Two experiment runs are made by switching on and off the ozone radiative feedback for the non-interactive and the interactive control runs, respectively. The QBO period is prolonged from 27 to 48 months in the switched-on run, while it is shortened from 31 to 20 months in the switched off run, demonstrating that the interactive ozone does prolong the QBO period."

In Shibata and Deushi (2005), they showed that the QBO periods simulated with interactive ozone are longer than those simulated with non-interactive ozone. However, their short report in Geophys. Res. Lett. did not mean to be comprehensive or exhaustive. For example, we don't know whether the climatologies of interactively generated ozone in Shibata and Deushi (2005) are close to those of the specified ozone in their non-interactive runs. Bushell et al. (2010) illustrated that the differences between the two ozone climatologies gave rise to differences in tropical upwelling between 100 and

> 4 hPa which led to a 12-month difference between the simulated mean QBO periods with the different ozone climatologies. We believe that many issues need to be explored to further our understand the discrepancies of the QBO periods between the non-interactive and interactive ozone experiments, which is beyond the scope of this paper.

> (2) Using a mechanistic model with a one-dimensional representation for mean flow and a three-dimensional depiction for Kelvin and Rossby-gravity waves, Cordero et al. (1998) demonstrated that ozone feedbacks lengthened the QBO period by about 2 months. Cordero and Nathan (2000) further developed a more sophisticated mechanistic model with a two-dimensional representation for mean flow and a three-dimensional depiction for Kelvin and Rossby-gravity waves, and surprisingly found that ozone feedbacks had little influence on the QBO period. We believe that more studies should be conducted to reconcile those discrepancies between the results from those two mechanistic models.

Also, the stratospheric cooling indirectly increases ozone because it reduces the ozone loss rate in the upper stratosphere, owing to the strong positive temperature dependence of the Chapman reactions and the NOx cycle on the ozone loss rate. So, I think that you should evaluate the influence of ozone on the QBO period, compared to the QBO shortening due to radiative damping.

> As mentioned above, we have added one paragraph (lines 405-415) to address this issue. However, some other interesting issues are beyond the scope of this paper and warrant further research.

Minor comments
L410-431. Redundant descriptions. You can omit most of them.

> This part was added to satisfy Reviewer 1 and Petr Šácha (who made a short comment during the Discussion stage). It is hard to make everyone happy.

**References**

Bushell, A. C., Jackson, D. R., Butchart, N., Hardiman, S. C., Hinton, T. J., Osprey, S. M., and Gray, L. J.: Sensitivity of GCM tropical middle atmosphere variability and climate to ozone and parameterized gravity wave changes, J. Geophys. Res., 115, D15101, https://doi.org/10.1029/2009JD013340, 2010.

Cordero, E. C., and Nathan, T. R.: The influence of wave- and zonal-mean ozone feedbacks on the quasi-biennial oscillation, J. Atmos. Sci., 57, 3426–3442, https://doi.org/10.1175/1520-0469(2000)057%3c3426:TIOWAZ%3e2.0.CO;2, 2000.

Cordero, E. C., Nathan, T. R. and Echols, R. S.: An analytical study of ozone feedbacks on Kelvin and Rossby–gravity waves: Effects on the QBO, *J. Atmos. Sci.*, 55, 1051–1062, https://doi.org/10.1175/1520-0469(1998)055<1051:AASOOF>2.0.CO;2, 1998.

Shibata, K., and Deushi, M.: Radiative effect of ozone on the quasi-biennial oscillation in the equatorial stratosphere, Geophys. Res. Lett., 32, L24802, https://doi.org/10.1029/2005GL023433, 2005.

---

## Author Response (AR3)

**Response to Editor**

**Editor Decision: Publish subject to minor revisions (review by editor) (22 Mar 2021) by Peter Haynes**

Comments to the Author:

You have made substantial changes in response to the comments of the two referees, so the paper is now closer to being suitable for publication. However, having looked at the revised paper carefully my view is that way in which the changes have been made is potentially confusing. I was considering sending the paper back to the referees for their comments, but have decided against that for the moment.

I have set out below the two aspects of the revision that I see as confusing. Please consider making appropriate changes or else provide responses arguing why no change is needed.

I hope to be able to accept the paper for publication at the next stage of the process, but to do so I need to feel that the referees' comments have been properly addressed, in a way that does not confuse the reader.

Thank you very much for your critical review. As a result, our paper has been thoroughly revised, and becomes clearer and more coherent. Hopefully, we have satisfactorily addressed all of the referees' concerns.

**(1) radiative damping**

Referee 1 made the point that the Plass (1956) paper which you originally used to motivate and quantify increased damping rates associated with increased $CO_2$ really says very little about change in damping rate. As the referee recommended -- and I think this is the approach you have taken. But you still make significant reference to Plass (1956) as you present the details of your approach.

The bottom line is that the idea of damping rate and ways to calculate is well-developed in the literature, including the idea that damping rate may change as a result of increased $CO_2$ -- see Fels (1985) for example. Bringing in Plass (1956) and then explaining why his results are not relevant is a complete distraction. If you want to acknowledge to the contribution of Plass (1956) in, for example, recognising that the radiative properties of the stratosphere will change with increased $CO_2$ then do that in the introduction -- though his was one of several contributions on effects of increasing $CO_2$, with the paper of Manabe and Wetherald (1967) being the most frequently cited (appropriate in my view if one takes account various important innovations in their calculation).

I have given 3 examples below where the text needs attention in this respect.
- L128-133: "Given the fact that the QBO period is influenced by the radiative damping (Plumb 1977; Hamilton 1981), a natural question to ask is whether it could play a role on the trend of the QBO in a warming climate. Plass (1956) showed that when the CO2 concentration is increased from 330 ppmv to 660 ppmv, the cooling rate increases significantly in the middle and upper stratosphere while it is not changed below the 24 km height level. The cooling rate is increased by about 50% around the 40 km height level (see his Figure 8)."

  If we agree that Plass's calculation tells us little about damping rate -- then the above sentences are very confusing -- most reader will assume that there is a connection between 'radiative damping' in the first sentence and 'cooling rate' in the second.
- L207-216: "Since the cooling rate is increased by about 50% around the 40 km height level (Plass, 1956), the radiative damping rates are expected to also increase in the middle and upper stratosphere as the CO2 concentration rises. However, the relative change of cooling rate $Q$ in response to the increasing CO2 is not identical to that of Newtonian cooling ..."

Again the Plass has very little useful relation to the proper calculation of damping rate, so what is the point in referring to Plass (1956) and then saying in the following sentence that a different calculation to that in Plass (1956) is needed ...?

- L222-226: "In FIG. 1b the black line depicts the ratio for the broadband longwave radiation (5 $\mu m$ −100 $\mu m$) and the red line delineates the ratio for the CO2 absorption band (12 $\mu m$ − 18 $\mu m$) used by Plass (1956). For the CO2 absorption band, the calculated ratio is evidently comparable to the ratio of cooling rates between the doubled CO2 and the reference CO2 shown in figure 8 of Plass (1956), with an additional small increase (<1.1) below the 24 km level."

Again, what is shown in your Figure has very little to do with Plass (1956) -- his Figure 6 shows cooling rate -- calculated on the basis of a fixed temperature profile and various CO2 concentrations. An essential part of your calculation is the heating/cooling rate difference arising from a temperature difference. In trying to bring in Plass (1956) here you are actually undermining the credibility of your own (updated) calculation.

We agree with you that Plass (1956) is a red herring. Accordingly, it has been removed.

**(2) Effect of ozone**

Referee 2 suggested that the effect of changing ozone on the QBO might well be as important as that of changing damping rates due to increasing CO2.

You don't mention the possibility of an ozone effect on the QBO until Section 4. (You mention the effect of the QBO ozone earlier, but that is different.) Then suddenly you give L344-354, which focuses only on the effect of ozone on the thermal damping rate, via its long-wave effect -- and essentially dismisses any role for ozone, when your reply to the referee has seemed to say that ozone is complicated and the resolving the role of ozone is not within the scope of your paper.

In order for the reader not to be confused by this there needs to be some introductory sentence somewhere which introduces the idea that changes in ozone may be important and furthermore it needs to be acknowledged that changing ozone may affect the QBO in other ways -- for example the Hasebe (1994) paper focuses on the short-wave heating role of ozone. As with your paper as a whole, you are in any case considering only one part the QBO mechanism -- the radiative damping of planetary-scale waves -- when there are many other parts of the QBO mechanism that may change as a result of increasing greenhouse gases -- and that includes parts that involve changes in ozone. You acknowledge this point in a fairly general way in the final sentence of your abstract and more clearly in the final sentence of the paper. Your paper will be more, not less, valuable if you make this point as clear as possible.

We have revised this part significantly and have tried our best to make it as clear as possible.